# SPOGW: A SCORE-BASED PREFERENCE OPTIMIZA-
# TION METHOD VIA GROUP-WISE COMPARISON FOR
# WORKFLOWS

## ABSTRACT

Large language models (LLMs) have exhibited significant capabilities in address-
ing challenging problems throughout various fields, often through the use of agen-
tic workflows that adhere to structured instructions and multi-step procedures.
However, designing such workflows demands substantial manual effort, posing
challenges to scalability and generalizability. Recent studies have aimed to min-
imize the human intervention needed for their construction, leading to advances
in automated techniques for optimizing agentic workflows. However, current ap-
proaches are often constrained by their limited representational capacity, insuf-
ficient adaptability, weak scalability, and pairwise comparison paradigm—issues
that stem primarily from a dependence on discrete optimization techniques. To
overcome these limitations, we introduce a new score-based preference approach,
refereed as SPOGW, which operates directly on cardinal reward signals through
group-wise comparison and enables more efficient and stable optimization in a
continuous space. SPOGW incorporates Iterative Offline GRPO (ioGRPO) with
advantage-masked KL restriction (mKL), which regulates training update by plac-
ing greater emphasis on the advantageous regions of the policy response. In five
benchmark datasets covering mathematical reasoning, coding, and question an-
swering, SPOGW matches or exceeds the performance of current state-of-the-art
approaches, presenting a viable and forward-looking methodology for automated
generation and optimization of agentic workflows.

## 1 INTRODUCTION

Large Language Models (LLMs) have demonstrated versatile capabilities in addressing challenging
tasks across numerous domains, such as data interpretation, code generation, mathematical problem
solving, and question answering (Liu et al., 2024a), (Li et al., 2024a), (Zhong et al., 2024), (Wang
et al., 2024), (Xu et al., 2023). However, the progress of LLM-based systems is considerably depen-
dent on hand-crafted agentic workflows—predefined sequences of LLM calls coupled with precise
instructions. The substantial human effort involved in developing and refining these workflows im-
pedes scalability, restricts adaptability to novel or intricate scenarios, and complicates knowledge
transfer between different tasks (Tang et al., 2023).

A key research direction that has thus gained traction aims to overcome the constraints of static work-
flows through automated techniques for systematically generating and refining workflows. Such
optimizations may be applied at multiple levels, such as improving prompts, adjusting hyperparam-
eters, or redesigning the workflow architecture itself (Chen et al., 2023), (Hu et al., 2024), (Song
et al., 2024), (Zhang et al., 2024b), (Li et al., 2024b), (Zhang et al., 2024a).

Current automated optimization techniques are often limited by predefined structural templates and
inflexible representations of the workflow space (Khattab et al., 2024), (Liu et al., 2024b), (Yuk-
sekgonul et al., 2024), (Zhuge et al., 2023). For instance, while DyLAN (Liu et al., 2024b) delib-
erately designs the communication protocol for LLM-based debates, it does not explore alternative
interaction patterns. GPTSwarm (Zhuge et al., 2023) utilizes graph representations and applies rein-
forcement fine-tuning for improvement, yet its failure to account for conditional states within graphs
inherently constrains the explorable solution space.

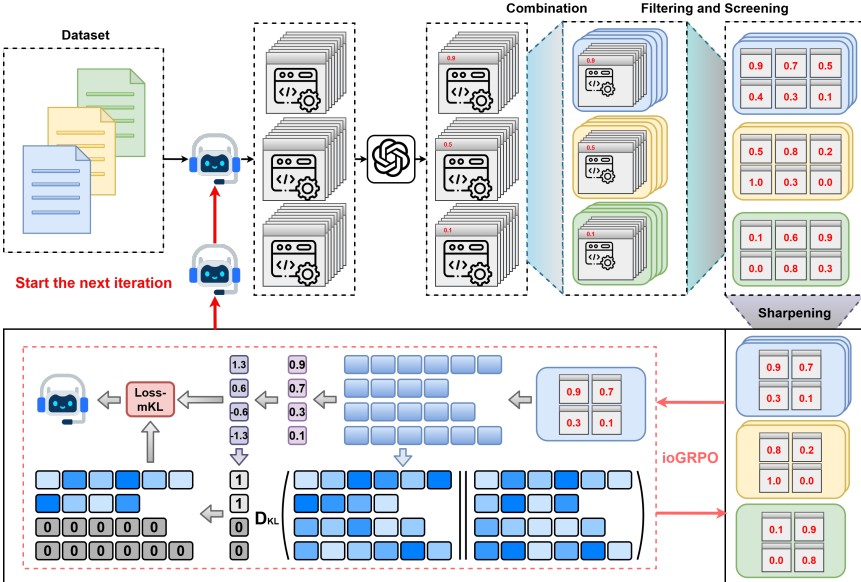

Figure 1: **Pipeline of SPOGW:** The framework generates multiple workflows for each query, then executes and evaluates each workflow to obtain a score, and then conducts combination and subsequent group-wise data processing, which feeds into ioGRPO optimization cycle.

To enhance the expressiveness and adaptability of workflow representations, approaches like ADAS (Hu et al., 2024), Aflow (Zhang et al., 2024b), and ScoreFlow (Wang et al., 2025) utilize workflow representations that are based on code. However, ADAS is hampered by the accumulation of irrelevant data and the increasing complexity during optimization, ultimately reducing its performance. Aflow improves the representation of workflows via code by incorporating a core element known as the named node, which encapsulates settings for LLM invocations to enable detailed modeling. The method also includes dedicated operators that carry out predefined logic for composing nodes. However, the effectiveness of Aflow's optimization based on Monte Carlo Tree Search is constrained by early convergence, and its discrete nature hinders scalability. ScoreFlow incorporates the Direct Preference Optimization (DPO) (Rafailov et al., 2023) RL technique into workflow optimization and adapts it to incorporate quantitative feedback. However, its optimization framework is severely constrained by a strong dependence on pairwise preference data, leading to rigidity. Essentially, it requires reframing performance assessment as a binary comparison process instead of directly optimizing a continuous metric of performance, thus hindering its ability to inherently integrate cardinal reward signals.

To address these challenges, we introduce SPOGW, a Score-based Preference Optimization method via Group-Wise comparison. SPOGW directly leverages cardinal reward signals and conducts optimization in a continuous space, thereby overcoming the inherent limitations of traditional pairwise preference paradigms. Our approach is built upon two key components:

- **Iterative Offline GRPO (ioGRPO)**, which decouples data collection from policy updates. By performing offline sampling and reward acquisition before training, ioGRPO eliminates the instability caused by on-the-fly code execution and API calls during optimization. The process runs in iterative cycles, where each iteration uses the previous checkpoint as the both old and reference policy for generating new training data.
- **Advantage-Masked KL Restriction (mKL)**, which selectively applies KL restriction penalties only to advantageous responses (those with positive advantage values). This ensures the policy stays aligned with high-quality behaviors from the reference model while avoiding unnecessary constraints from low-quality outputs.

Together, these innovations enable more stable, efficient, and scalable optimization of agentic workflows. Experiments across mathematical reasoning, coding, and QA benchmarks show that SPOGW matches or surpasses state-of-the-art methods, demonstrating its effectiveness as a general-purpose framework for automated workflow generation and optimization.

## 2 RELATED WORKS

### 2.1 REINFORCEMENT LEARNING FOR ADVANCED REASONING IN LLMS

The integration of reinforcement learning (RL) to improve LLM reasoning has attracted considerable attention (Cheng et al., 2025), (Zhang et al., 2025), (Xiong et al., 2025), owing to its ability to enable self-improvement without dependence on human-annotated solutions. This is commonly realized through fine-tuning on sophisticated reasoning problems, with the objective of promoting varied reasoning strategies (Gandhi et al., 2025), (Yue et al., 2025). Notable advances such as OpenAI o1 (Jaech et al., 2024) and DeepSeek-R1 (Guo et al., 2025) illustrate that RL methods can be effectively applied to large-scale commercial systems, substantially pushing the boundaries of reasoning performance and unveiling emergent skills including extended reasoning chains. Recent progress has employed reinforcement learning guided by scalar feedback signals (Jaech et al., 2024), (Guo et al., 2025), (Liu et al., 2025), (Yu et al., 2025). For instance, a positive reward (e.g., +1) may be assigned for accurate outputs, while a negative penalty (e.g., –1) is applied to erroneous responses to provide clear learning signals. Common algorithmic choices in this line of work include online policy optimization techniques such as REINFORCE (Williams, 1992), Proximal Policy Optimization (PPO) (Schulman et al., 2017), Group Relative Policy Optimization (GRPO) (Shao et al., 2024), and Decoupled Clip and Dynamic Sampling Policy Optimization (DAPO) (Yu et al., 2025). Although powerful within their respective specialized domains, current research on applying RL algorithms to workflow optimization techniques remains relatively scarce. Furthermore, the adaptability and effectiveness of existing RL algorithms for workflow optimization tasks lack sufficient empirical validation and analysis.

### 2.2 AUTOMATED AGENTIC WORKFLOW OPTIMIZATION

**Agentic Workflows** Agentic workflows and autonomous agents (Zhuge et al., 2023), (Hong et al., 2024), (Wang et al., 2023a) constitute two principal paradigms for applying LLMs. The former operate through fixed, predefined sequences of steps—orchestrating multiple calls to LLMs—to accomplish tasks in a structured manner. In contrast, the latter tackle problems adaptively via self-directed reasoning and action. Unlike autonomous agents, which often demand carefully crafted action spaces and decision rules tailored to particular environments, agentic workflows can be built upon accumulated human expertise and refined through iteration. This positions them as a more automatable and scalable approach for many practical applications.

**Automated Workflow Optimization** Workflow optimization techniques (Zhou et al., 2024), (Hu et al., 2024), (Zhang et al., 2024b), (Li et al., 2024b), (Zhang et al., 2024a), (Wang et al., 2025) aim to improve the structural design of workflows, enhancing their robustness across varied tasks. However, their effectiveness is often constrained by rigid representations—for instance, the loss of conditional logic in graph-based structures—which narrows the search space and limits adaptability to complex scenarios. To overcome these limitations, methods such as ADAS (Hu et al., 2024), Aflow (Zhang et al., 2024b) and ScoreFlow (Wang et al., 2025) employ code-based workflow representations. Aflow enhances code-based workflow representation by introducing a foundational component called named node, which packages parameters for LLM calls to support fine-grained workflow modeling. However, the efficacy of Aflow's Monte Carlo Tree Search-based optimization is limited by premature convergence, while its discrete optimization property impedes scalability. ScoreFlow integrates the Direct Preference Optimization (DPO) (Rafailov et al., 2023) reinforcement learning method into workflow optimization and extends it to account for quantitative feedback. However, its optimization paradigm is critically limited by its heavy reliance on pairwise preference data, resulting in inflexibility.

## 3 METHODS

### 3.1 SCORE-BASED PREFERENCE DATASET

**Data construction** In the score-based workflow application, for each query $q$ in the dataset $\mathcal{D}$, a generator LLM produces $m$ corresponding workflows (denoted as $g_i(q), i \in [m]$). Executing each workflow $g_i(q)$ yields a result for each query $q$. Subsequently, these results are evaluated to

produce the corresponding scores $s_i$ ($s_i \in [0,1]$). In experiments, the workflows are executed by independently querying an executor LLM; Unlike self-improvement methods (Jiang et al., 2024) that employ the generator model for evaluation, score-based workflow (Wang et al., 2025) leverages external resources (e.g., a validation dataset and an executor LLM) to realize the self-referential property of the iterative process. However, ScoreFlow (Wang et al., 2025) is limited by the pairwise comparison paradigm, which constructs score-based workflows into preference pairs, thereby restricting the scale of comparison samples and lacking flexibility and scalability.

Thus, we construct the group-wise training data set for query $q$ from a new perspective, with a initial group size of $n$ ($n \le m$) pairs. For each data instance, we define

$$D_q = \{q, (g_1(q), s_1), \ldots, (g_n(q), s_n)\}\}, \tag{1}$$

where consists of a query $q$, the corresponding workflows $g_j(q)$, and their respective scores $s_j$, $j \in [n]$. Since $m$ workflows are initially generated for each query, a total of $M$ group-wise training instances can be created for each query $q$ through combinations, i.e., $M = C(m, n)$, which represents the number of ways to choose n distinct elements from a set of m elements. Thus, we obtain a group training set for query $q$, i.e., $\mathcal{D}_q = \{D_q^1, \ldots, D_q^M\}$. Finally, the complete preprocessed group-wise training dataset $\mathcal{D}_{\text{pre}}$ is formed by aggregating the data from all queries, i.e., $\mathcal{D}_{\text{pre}} = \bigcup_{q \in \mathcal{D}} D_q$.

**Filtering and Screening**   The quality of the initial group-wise dataset $\mathcal{D}_{\text{pre}}$ is heterogeneous, with significant variation across samples. Certain instances suffer from highly similar sampled responses and nearly identical intra-group reward scores. Utilizing these suboptimal samples would detrimentally impact the efficacy of advantage estimation. Therefore, we design a subsequent data curation pipeline to post-process the raw dataset. The objective of this pipeline is to yield a refined dataset characterized by superior *intra-group diversity* and clearer *distinction between high- and low-quality responses*. This high-quality data enables the advantage calculation to produce a stronger and more unambiguous learning signal, thereby enhancing the efficiency of the reinforcement learning process and ultimately leading to improved final performance.

Specifically, the filtering step checks whether all workflows in a group have identical scores. If so, the entire instance is discarded because it provides no meaningful signal for advantage estimation. Otherwise, the instance is retained for further processing. In actual implementation, the generation of combinations is combined with filtering, where each generated instance is examined immediately to decide whether to discard it. The screening process is applied to this filtered set. For each remaining instance, the variance of the reward scores $\text{Var}(s_1, ..., s_n)$ for its $n$ responses is computed. The instances are then ranked in descending order based on this variance. We select the top-$N$ instances with the highest variance for inclusion in the final training set (or all instances if the total number is less than $N$). A high variance indicates a reward distribution with sufficient distinction among responses, enabling the model to more effectively learn nuanced preferences. This screening procedure ensures *intra-group diversity* and mitigates the risk of advantage calculation failure due to reward homogenization.

**Group Sharpening**   To achieve a higher effective variance and a more polarized reward distribution using fewer samples, we perform a curated step on the screened dataset, called *group sharpening*. For a given data instance $D_q$, we first sort the responses and their corresponding rewards in ascending order based on the reward value, resulting in the same data instance but under different orders, i.e., $\hat{D}_q = \{q, (g_{(1)}(q), s_{(1)}), \ldots, (g_{(n)}(q), s_{(n)})\}\}$, where $s_{(1)} \le ... \le s_{(n)}$. The sharpening operation then retains only the top-$t$ and bottom-$t$ responses, effectively constructing a new, sharper instance:

$$\tilde{D}_q = \{q, (g_{(1)}(q), s_{(1)}), \ldots, (g_{(t)}(q), s_{(t)}), (g_{(n-t+1)}(q), s_{(n-t+1)}), \ldots, (g_{(n)}(q), s_{(n)})\}\}, \tag{2}$$

where $2t < n$. The group size is thus reduced from $n$ to $2t$. By focusing on the most positively and negatively rewarded examples, this technique amplifies the contrast within the data, yielding a stronger and clearer learning signal for the advantage estimator. This leads to more stable and efficient policy updates during training.

### 3.2 FROM GRPO TO ITERATIVE OFFLINE GRPO

Group Relative Policy Optimization (GRPO) is an online reinforcement learning algorithm commonly employed for fine-tuning Large Language Models (LLMs). It extends the framework of

Proximal Policy Optimization (PPO) (Schulman et al., 2017), while avoiding the requirement for explicit value function estimation by computing advantages through the comparative performance of grouped actions. GRPO achieves more stable and robust learning through relative comparisons within a group of samples. By focusing on the relative advantage of each action compared to others in the group, GRPO effectively circumvents the challenges associated with value function estimation bias. In the setting of LLM policy optimization, consider a model policy with parameters $\theta$. For every query $q$ from a set $\mathcal{Q}$, a set of candidate responses $\{y_i\}_{i=1}^n$ is generated under the previous policy $\pi_{\text{old}}$. These samples are then evaluated by a reward model, producing a corresponding set of rewards $\{R_i\}_{i=1}^n$. The objective function for GRPO is expressed as:

$$\mathcal{J}_{\text{GRPO}}(\theta) = \frac{1}{n} \sum_{i=1}^n \frac{1}{|y_i|} \sum_{t=1}^{|y_i|} \left\{ \min\left[ r_{i,t}(\theta)\hat{A}_{i,t}, \text{clip}(r_{i,t}(\theta), 1-\epsilon, 1+\epsilon)\hat{A}_{i,t} \right] \right\}, \tag{3}$$

where the probability ratio $r_{i,t}(\theta)$ is defined as the relative probability of generating a response under the current policy $\pi_\theta$ compared to the old policy $\pi_{\text{old}}$ under which the responses were initially sampled: $r_{i,t}(\theta) = \frac{\pi_\theta(y_{i,t}|q,y_{i,<t})}{\pi_{\text{old}}(y_{i,t}|q,y_{i,<t})}$. Here, $\epsilon$ represent hyperparameters. The advantage value $\hat{A}_{i,t}$ is computed for all tokens within a response by standardizing the rewards $\{R_i\}_{i=1}^n$—specifically, by subtracting the group mean and dividing by the group standard deviation. $\hat{A}_{i,t} = \frac{R_i - \mu}{\sigma}$ where $\mu$ and $\sigma$ are the mean and standard deviation of the rewards within the group, respectively.

However, within the workflow optimization process, scoring an individual workflow involves code execution and potentially unstable API calls. Employing the original GRPO methodology, which requires generating and evaluating workflows on-the-fly during training followed immediately by gradient updates, introduces a critical point of failure. If any single code execution or API call fails or hangs during a training step, the entire training process is forced to halt. Consequently, the stability of each individual code run and API invocation directly impacts the overall stability of the training procedure, rendering this approach infeasible for practical implementation.

To circumvent these challenges, we propose a variant of GRPO, termed **Iterative Offline GRPO (ioGRPO)**. This method modifies the forward pass of the original GRPO algorithm by decoupling data collection from policy optimization. Specifically, response sampling and reward acquisition are performed as a separate, offline pre-processing step *before* the commencement of training. During the actual training phase, the optimization process directly reads from a pre-collected dataset containing queries $q$, their corresponding sets of responses $\{g_1, ..., g_n\}$, and associated rewards $\{s_1, ..., s_n\}$ to compute the policy gradient loss. Furthermore, starting from a base model, we conduct multiple iterative cycles. After each training iteration, a new model checkpoint is saved. This checkpoint serves a dual purpose: 1) it becomes the starting point for the next training iteration, and 2) it acts as the old policy $\pi_{\text{old}}$ for the subsequent round of data collection. This checkpoint is then used to re-sample a new set of responses and acquire their corresponding rewards, refreshing the training dataset for the next iteration. This iterative process effectively decomposes the monolithic GRPO training procedure into two distinct, alternating phases: **dataset collection** and **policy update**. This separation successfully eliminates the adverse effects of code execution and API instability on training robustness while simultaneously achieving a significant reduction in overall training time.

We note that our Iterative Offline GRPO (ioGRPO) is not a pure offline reinforcement learning method. While each training epoch uses a static, offline dataset, the overall process involves iterative data collection from the environment after each epoch, where the new policy interacts with the problem dataset to generate fresh training data. This hybrid approach combines the stability of offline training with the adaptability of online data collection, specifically designed to handle the instability in workflow execution. In Appendix C.1, we have conducted a more detailed discussion and experimental analysis on this.

### 3.3 ADVANTAGE-MASKED KL RESTRICTION

According to recent research efforts, including (Yu et al., 2025), the distribution of long-chain reasoning models can undergo substantial divergence from the initial model during training, making such restriction unnecessary. However, when the reference model is chosen as the checkpoint from the preceding iteration—which is also the model that generated the offline training data for the current round—the role and effect of the KL restriction warrant further analysis. In the objective

function of the **ioGRPO**, we add the term $-\mathbb{D}_{\text{KL}}[\pi_\theta || \pi_{\text{ref}}]$, which discourages the updated policy from diverging too far from the original reference policy. the KL penalty term in the latter part of the expression can be formulated as:

$$\mathbb{D}_{\text{KL}}[\pi_\theta || \pi_{\text{ref}}] = \frac{\pi_{\text{ref}}(y_{i,t}|q, y_{i,<t})}{\pi_\theta(y_{i,t}|q, y_{i,<t})} - \log \frac{\pi_{\text{ref}}(y_{i,t}|q, y_{i,<t})}{\pi_\theta(y_{i,t}|q, y_{i,<t})} - 1 \qquad (4)$$

where $\pi_\theta$ denotes the new policy being trained and $\pi_{\text{ref}}$ represents the reference policy. This KL restriction term constrains the deviation of the new policy from the reference policy, ensuring the updated policy does not diverge excessively during iterative optimization. For the Iterative Offline GRPO framework, the reference policy $\pi_{\text{ref}}$ can be selected either as the initial base model or as the model checkpoint saved from the previous training iteration. Note that Eq.(4) adopts the KL restriction definition from the original GRPO paper (Shao et al., 2024), which uses an unbiased estimator to ensure positive values.

However, the original objective function applies the KL restriction uniformly to all responses for a given query. Within a pre-collected dataset, each group contains a mix of both high-quality (advantageous) and low-quality (disadvantageous) responses. Applying the restriction to advantageous responses is desirable, as it prevents the new policy from deviating excessively from the high-performing strategies of the reference model. Conversely, applying the same restriction to disadvantageous responses would force the new policy to remain close to the reference model's poor strategies. This latter effect is counterproductive and misaligned with the core objective of reinforcing the generation of high-advantage outputs.

Our key modification involves linking the advantage values, computed during the ioGRPO objective estimation, directly to the KL restriction. This integration imbues the KL penalty term with an *advantage-aware selectivity*. Specifically, for a given training sample $\tilde{D}_q$ defined in Eq.(2), the advantage value $A_i$ is computed for each response $g_i(q)$, $i \in [n]$. A positive $A_i$ indicates that the corresponding response should be reinforced, whereas a negative $A_i$ suggests that it should not. Based on this intuition, we introduce a **Advantage-Masked KL Restriction (mKL)** $m_i$, defined for each response in the sample as:

$$m_i = \mathbb{I}(A_i > 0), \qquad (5)$$

where $\mathbb{I}$ is the indicator function, $1 \leq i \leq n$. The purpose of this mask is to filter the $n$ sampled responses for a given query $q$, selecting only the $l$ ($l < n$) *advantageous responses* ($A_i > 0$) for inclusion in the KL penalty calculation. The KL restriction thus only applies to these advantageous responses, effectively ignoring the contributions from the disadvantageous ones. This mechanism ensures that the KL penalty term constrains the new policy $\pi_\theta$ towards the *advantageous segments* of the reference policy $\pi_{\text{ref}}$, rather than constraining it against the entirety of $\pi_{\text{ref}}$'s output distribution. The modified GRPO objective function, incorporating the proposed **mKL**, is therefore given by:

$$\mathcal{L}_{\text{ioGRPO-mKL}}(\theta)$$
$$= \frac{1}{n} \sum_{i=1}^{n} \frac{1}{|y_i|} \sum_{t=1}^{|y_i|} \left\{ \min \left[ r_{i,t}(\theta)\hat{A}_{i,t}, \text{clip}(r_{i,t}(\theta), 1-\epsilon, 1+\epsilon)\hat{A}_{i,t} \right] - \beta \cdot m_i \cdot \mathbb{D}_{\text{KL}}[\pi_\theta || \pi_{\text{ref}}] \right\}, \quad (6)$$

where $m_i$ is the mask value defined in Eq. (5), and $\beta$ is a scaling hyperparameter for the penalty term.

## 3.4 DISCUSSION ON GENERALIZABILITY

While our framework was designed for agentic workflow optimization, its core components—Iterative Offline GRPO (ioGRPO) and Advantage-Masked KL Restriction (mKL)—are not domain-specific. The modifications address fundamental challenges in adapting GRPO to domains where reward scoring is complex and unstable, such as when execution verification is required before simply direct scoring.

In principle, ioGRPO can be applied to classic tasks like mathematical reasoning and question answering, where data collection would involve sampling followed by direct scoring. The advantage-masked KL restriction provides a general mechanism for selectively constraining policy divergence based on advantage signals. Although empirical validation across diverse domains requires further

investigation, our framework offers a potentially viable solution for problem domains sharing similar characteristics with workflow optimization, particularly those with complex and unstable reward estimation processes.

# 4 EXPERIMENTS

## 4.1 EXPERIMENTAL SETUP

**Datasets** We center our evaluation on five publicly available datasets spanning diverse domains such as code generation, mathematics, and question answering. In particular, we employ the entire collections of HumanEval (Chen et al., 2021) and MBPP (Austin et al., 2021). To focus on advanced and complex problems within the MATH dataset, we extract level-5 difficulty questions from the following categories: Combinatorics and Probability, Number Theory, Pre-algebra, and Pre-calculus, mirroring the selection process of (Hong et al., 2024). For DROP (Dua et al., 2019) and HotpotQA (Yang et al., 2018), we adhere to the sampling protocols established in (Hu et al., 2024), (Shinn et al., 2023), (Zhang et al., 2024b), and (Wang et al., 2025), randomly drawing 1,000 instances from each. These samples are then partitioned into training sets and test sets using a 1:4 ratio.

**Baselines** Our evaluation includes several manually constructed static workflow baselines: direct LLM calls, Chain of Thought (Wei et al., 2022), Self-Consistency CoT (ensembling 5 generated responses) (Wang et al., 2022), MedPrompt (3 responses with 5 votes) (Nori et al., 2023), MultiPersona Debate (Wang et al., 2023b), and Self-Refine (executed over 2 rounds) (Madaan et al., 2023). We also include comparisons with state-of-the-art automated workflow optimization techniques based on code representations: ADAS (Hu et al., 2024) , Aflow (Zhang et al., 2024b) , and ScoreFlow (Wang et al., 2025). The first two methods both utilize `GPT-4o-mini-2024-07-18` as their underlying optimization model. Following the configuration used in (Zhang et al., 2024b), the number of iteration rounds for Aflow is set to 20. For ScoreFlow, `Qwen2.5-7B-Instruct` is employed as the generator and `GPT-4o-mini-2024-07-18` as the executor, with the iteration round set to 3. (Unlike non-training methods that directly use powerful models for optimization, ScoreFlow involves training an extra open-source model.)

**Models** In our primary setup, similarly, `Qwen2.5-7B-Instruct` (Yang et al., 2025) serves as the foundational generator model, with inference conducted using vLLM (Kwon et al., 2023). The executor is `GPT-4o-mini-2024-07-18`, accessible via API with a temperature set to 0. For ablation experiments, we substitute the generator with `Qwen2.5-3B-Instruct` and `Qwen2.5-14B-Instruct`, while still keeping `GPT-4o-mini-2024-07-18` as the executor. All experiments are conducted on four H20 GPUs employing LoRA (Hu et al., 2022) for efficient fine-tuning.

**Evaluation Metrics** The final performance is measured by task solve rates, averaged over 3 independent evaluation runs. To mitigate formatting inconsistencies across outputs, `GPT-4o-mini-2024-07-18` serves as the judge model for the MATH, DROP, and HotpotQA datasets. During each of the 3 optimization iterations, we generate 16 candidate workflows per problem (i.e. $m = 16$) and compute their scores. Meanwhile, the initial group size is set to 14 (i.e. $n = 14$), and the final group size (after group sharpening) is set to 8 (i.e. $t = 4$). To limit computational expense, no dedicated judge model is employed at this stage. Evaluation relies on F1 scores for DROP and HotpotQA, and on solve rates—also averaged over three runs—for all other benchmarks.

## 4.2 MAIN RESULTS

We present the main experimental results comparing SPOGW against a comprehensive set of baseline methods across five benchmark datasets spanning mathematical reasoning, coding, and question answering domains. As shown in Table 1, SPOGW consistently achieves state-of-the-art performance, outperforming all baseline methods on every benchmark.

Specifically, In **mathematical reasoning (MATH)**, SPOGW attains a solve rate of **62.3**%, surpassing the previous best method, ScoreFlow, by **2.3** percentage points. This demonstrates SPOGW's effectiveness in handling complex, multi-step reasoning tasks requiring structured problem-solving

Table 1: Main experimental results comparing SPOGW with baseline methods across five benchmarks: MATH (math reasoning), HumanEval and MBPP (coding), HotpotQA and DROP (question answering). SPOGW achieves state-of-the-art performance on all tasks, with improvements over the previous best method (ScoreFlow) indicated by up arrows.

| Methods | Math Reasoning | Coding | | Question Answering | | AVG |
|---|---|---|---|---|---|---|
| | MATH | HumanEval | MBPP | HotpotQA | DROP | |
| IO | 52.2 | 90.1 | 69.5 | 73.6 | 81.6 | 73.4 |
| CoT (Wei et al., 2022) | 53.4 | 91.6 | 70.4 | 73.4 | 83.2 | 74.4 |
| CoT SC (Wang et al., 2022) | 53.8 | 92.9 | 71.3 | 74.0 | 83.2 | 75.0 |
| MedPrompt (Nori et al., 2023) | 53.7 | 92.1 | 69.2 | 74.4 | 83.0 | 74.5 |
| MultiPersona (Wang et al., 2023b) | 51.9 | 92.9 | 70.4 | 73.1 | 81.3 | 73.9 |
| Self Refine (Madaan et al., 2023) | 50.0 | 91.1 | 70.0 | 73.6 | 82.5 | 73.4 |
| ADAS (Hu et al., 2024) | 51.7 | 88.8 | 68.7 | 78.5 | 81.3 | 73.8 |
| Aflow (Zhang et al., 2024b) | 55.8 | 92.9 | 82.9 | 77.9 | 83.5 | 78.6 |
| ScoreFlow (Wang et al., 2025) | 60.0 | 95.1 | 83.2 | 84.1 | 84.3 | 81.3 |
| **SPOGW(Ours)** | **62.3**↑2.3 | **96.2**↑1.1 | **84.1**↑0.9 | **85.0**↑0.9 | **85.3**↑1.0 | **82.6**↑1.3 |

Table 2: Performance comparison of different generator models with and without SPOGW optimization on HumanEval and HotpotQA. SPOGW enables smaller models to achieve performance competitive with larger baseline models.

| Generator Model | HumanEval | HotpotQA |
|---|---|---|
| Qwen2.5-3B-Instruct | 91.9 | 84.1 |
| Qwen2.5-7B-Instruct | 93.4 | 84.2 |
| Qwen2.5-14B-Instruct | 94.4 | 84.7 |
| **Qwen2.5-3B-Instruct-SPOGW** | **94.1**↑2.2 | **84.3**↑0.2 |
| **Qwen2.5-7B-Instruct-SPOGW** | **96.2**↑2.8 | **85.0**↑0.8 |

workflows. For **code generation tasks,** SPOGW achieves **96.2**% on HumanEval and **84.1**% on MBPP, exceeding ScoreFlow by **1.1** and **0.9** percentage points, respectively. This improvement highlights SPOGW's capability in generating functionally correct code through optimized workflow structures. In **question answering**, SPOGW obtains **85.0**% on HotpotQA and **85.3**% on DROP, representing gains of **0.9** and **1.0** percentage points over ScoreFlow. These results indicate that SPOGW effectively handles multi-hop reasoning and discrete reasoning over textual content.

Across all benchmarks, SPOGW achieves an average performance of **82.6**%, a **1.3** percentage point improvement over the previous state-of-the-art. Notably, SPOGW not only outperforms automated workflow optimization methods (ADAS, Aflow, ScoreFlow) but also exceeds carefully designed manual workflows such as MedPrompt, MultiPersona, and Self-Refine. The consistent superiority of SPOGW across diverse domains underscores the effectiveness of our group-wise preference optimization approach. The improvements are particularly significant in mathematical reasoning, where the structured nature of workflows plays a crucial role in solving complex problems. These results validate SPOGW as a robust and general-purpose framework for automated workflow generation and optimization.

## 4.3 ABLATION STUDIES

**Analysis of the generator model** As shown in Table 2, we investigate the impact of the generator model by comparing Qwen2.5 models of varying sizes on HumanEval and HotpotQA. The results show that SPOGW optimization not only improves the performance of the original model but even effectively compensates for the limitations of the model scale: while baseline performance improves with increasing model size (e.g., HumanEval scores rising from 91.9 to 94.4), SPOGW-enhanced smaller models achieve performance comparable to or even surpassing larger baseline models. Specifically, Qwen2.5-3B-Instruct-SPOGW attains 94.1 on HumanEval, closely approaching the baseline 14B model's 94.4, while Qwen2.5-7B-Instruct-SPOGW reaches 96.2, exceeding all baseline models including the 14B variant. This indicates that SPOGW's group-wise optimization

Table 3: Ablation study on KL restriction configurations. Results show that combining the iterative checkpoint with selective KL mask yields the highest performance.

| Objective Function | Reference Model | Enable KL Mask | HumanEval |
|---|---|---|---|
| w/o KL term | None | ✗ | 94.9 |
| w/ KL term | Initial base model | ✗ | 94.4 |
| w/ KL term | Previous iteration's checkpoint | ✗ | 95.4 |
| **w/ KL term** | **Previous iteration's checkpoint** | ✓ | **96.2**↑0.8 |

Table 4: Impact of data processing methods on HumanEval performance. Progressive refinement from random sampling to screening and sharpening demonstrates the importance of data quality curation for effective policy optimization, with the combined approach yielding the optimal result.

| Dataset | Data Processing Methods | Filtering First | HumanEval |
|---|---|---|---|
| $\mathcal{D}_{RS}$ | Random Sampling | ✗ | 93.4 |
| $\mathcal{D}_{RSF}$ | Random Sampling | ✓ | 94.9 |
| $\mathcal{D}_{S}$ | Only Screening | ✓ | 95.7 |
| $\mathcal{D}_{SS}$ | **Screening and Sharpening** | ✓ | **96.2** |

effectively amplifies the capabilities of smaller models, reducing dependency on model scale while maintaining strong performance across reasoning tasks.

**Analysis of the KL restriction** As shown in Table 3, we ablate the impact of the KL restriction and the proposed advantage-masked mechanism on HumanEval performance. Removing the KL term entirely yields a score of 94.9, while applying KL regularization with the initial base model as reference degrades performance to 94.4, indicating that rigid constraint towards an outdated policy can hinder optimization. Switching the reference model to the previous iteration's checkpoint improves results to 95.4, demonstrating the benefit of iterative policy alignment. Finally, enabling the KL mask—which selectively applies KL penalty only to advantageous responses—further boosts performance to 96.2, underscoring that targeted restriction towards high-quality behaviors is crucial for stable and effective policy improvement.

**Analysis of the data processing method** As demonstrated in Table 4 and Figure 2, the progressive refinement of data processing methods significantly enhances model performance on HumanEval, with random sampling achieving 93.4, filtering improving to 94.9, screening alone reaching 95.7, and the combined screening and sharpening approach yielding the optimal 96.2. The variance and median interval length distributions reveal that Dataset 4 exhibits both higher variance and clearer separation among reward scores, confirming that our curated data processing pipeline effectively amplifies intra-group diversity and reward distinction, thereby providing stronger and clearer learning signals for advantage estimation and policy optimization.

**Analysis of the KL coefficient $\beta$, the group size, and the dataset size** Our ablation study on the KL coefficient ($\beta$), group size (2t), and dataset size (d) highlights the importance of balanced hyperparameters for stable and efficient policy optimization. As shown in Figures 3(a)–3(c), excessive KL regularization ($\beta = 0.2$) restricts exploration while insufficient regularization ($\beta = 0.025$) destabilizes learning, with the optimal $\beta = 0.1$ achieving the highest score (96.2). Similarly, SPOGW reaches peak performance at a group size of 2t = 8, where smaller or larger groups either reduce contrast for advantage estimation or introduce noise. Performance also follows an inverted U-shaped trend with respect to dataset size, peaking at d = 100 and declining for both smaller and larger datasets. These results collectively underscore the critical role of properly tuned regularization strength, group size, and data scale in maintaining sharp reward distinctions and preventing overfitting, thereby maximizing policy performance.

**Additional ablations and discussions** We conduct further ablation studies to validate our design choices. First, we compare ioGRPO with pure offline GRPO and a variant with intra-epoch interaction (ioGRPO (1/2 split)) on HumanEval (Appendix C.1). The results show that pure offline GRPO suffers from performance decay due to data shift, while ioGRPO maintains high performance. Second, we compare group-wise ioGRPO with pairwise Score-DPO on HumanEval (Appendix C.2); ioGRPO achieves 95.4 vs. 94.9 for Score-DPO, demonstrating the benefit of group-wise compar-

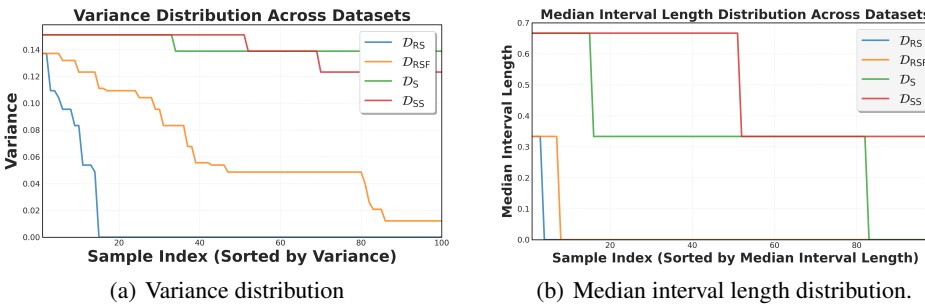

(a) Variance distribution

(b) Median interval length distribution.

Figure 2: Analysis of dataset characteristics under different processing methods shows that $\mathcal{D}_{SS}$ achieves superior variance and clearer quality separation. The training group size is fixed at 8. Median Interval Length (MIL) is the gap between the 4th and 5th highest scores.

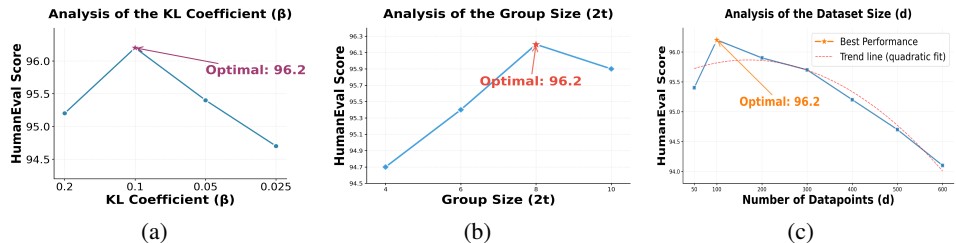

(a)                (b)                (c)

Figure 3: Analysis of the KL coefficient $\beta$, group size $2t$ and dataset size $d$ on HumanEval.

isons. Third, to assess evaluation robustness, we test SPOGW and ScoreFlow on HotpotQA using different judge models (Appendix C.3). SPOGW consistently outperforms ScoreFlow across judges, indicating no judge-specific overfitting. Additionally, training comparisons with ScoreFlow (Appendix B) show that SPOGW has similar optimization time but better performance and stability. We also analyze the computational cost of SPOGW to verify its cost-effectiveness, including API expenses for data collection and evaluation, comparison with baseline methods. The detailed cost breakdown is provided in Appendix D.

## 5 CONCLUSION

We present SPOGW, a score-based preference optimization method for automated agentic workflow generation that overcomes the limits of discrete optimization and pairwise comparisons via group-wise optimization in continuous space. SPOGW introduces three innovations: 1) variance-based group-wise data construction, 2) an Iterative Offline GRPO framework decoupling data collection from policy updates for stability, and 3) an advantage-masked KL restriction guiding policy divergence toward high-quality behaviors. Experiments on reasoning, coding, and QA benchmarks show SPOGW surpasses state-of-the-art methods, while ablations confirm each component's contribution and highlight optimal hyperparameter settings. SPOGW offers a scalable, effective framework that reduces manual design while maintaining strong performance across domains.

ETHICS STATEMENT

We ensure that our submission will not raise questions regarding the Code of Ethics.

REPRODUCIBILITY STATEMENT

This paper, together with the forthcoming supplementary materials, provides all necessary details for reproducing our results. Section 4 outlines the training methodology and data annotation standards, as well as the experimental setup, including hyperparameters and evaluation protocols. All datasets and baseline models used for comparison are publicly accessible on HuggingFace. Upon publication, we will also release the source code, training data, and model checkpoints for our data processing, training, and evaluation pipeline to support further research.

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

# A   DETAILS ABOUT THE ANALYSIS OF EACH HYPERPARAMETER

The blation studies highlight the importance of tuning key hyperparameters in SPOGW. An optimal KL coefficient ($\beta = 0.1$) anchors the policy to high-quality behaviors without stifling exploration. A balanced group size ($2t = 8$) maximizes reward contrast and stability, avoiding the drawbacks of too-small or too-large groups. Likewise, dataset size follows an inverted U-shape pattern, with $d = 100$ yielding the best performance and both smaller and larger datasets reducing efficiency. Together, these results demonstrate that carefully calibrated KL penalties, group sizes, and dataset scales are critical for stable and effective policy optimization. More details are in the following tables:

**Analysis of the KL coefficient $\beta$,**   The ablation study on the KL coefficient $\beta$ (Table 5 and Figure 3(a)) reveals a clear trade-off in the strength of the restriction: while excessive restriction ($\beta = 0.2$) limits exploration and produces 95.2, insufficient restriction ($\beta = 0.025$) leads to divergence of policy and reduces performance to 94.7. Optimal balance is achieved at $\beta = 0.1$, which maximizes performance at 96.2 by effectively anchoring the policy to high-quality behaviors without stifling improvement, demonstrating the critical role of tuned KL penalties in stable policy optimization.

**Analysis of the group size**   As evidenced in Table 6 and Figure 3(b), the group size parameter $2t$ demonstrates a clear optimal range for SPOGW performance on HumanEval, with $2t = 8$ achieving the peak score of 96.2, while smaller sizes ($2t = 4$, 94.7) lack sufficient contrast for effective advantage estimation and larger sizes ($2t = 10$, 95.9) introduce noise from medium-quality responses, confirming that a balanced group size is essential for maintaining sharp reward distinctions while providing adequate sample diversity for stable policy updates.

**Analysis of the dataset size**   As shown in Table 7 and Figure 3(c), the relationship between dataset size and performance follows an inverted U-shape pattern, with optimal results achieved at $d = 100$ (96.2 on HumanEval), while both smaller datasets ($d = 50$, 95.4) and larger datasets ($d \geq 200$, down to 94.1 at $d = 600$) yield inferior performance, indicating that while sufficient data are necessary for effective policy learning, excessive data points may introduce noise or overfitting that diminishes optimization efficiency.

Table 5: Ablation study on the KL coefficient $\beta$, showing performance on HumanEval. Optimal performance is achieved at $\beta = 0.1$, with larger or smaller values leading to degradation due to excessive restriction or insufficient alignment, respectively.

| KL Coefficient $\beta$ | HumanEval |
| --- | --- |
| $\beta = 0.2$ | 95.2 |
| $\beta = 0.1$ | **96.2** |
| $\beta = 0.05$ | 95.4 |
| $\beta = 0.025$ | 94.7 |

Table 6: Analysis of the group size $2t$ on HumanEval performance, showing a clear peak at $2t = 8$. The performance degradation at both smaller and larger sizes indicates the importance of balanced group construction for effective advantage estimation.

| Group Size $2t$ | HumanEval |
| --- | --- |
| $2t = 4$ | 94.7 |
| $2t = 6$ | 95.4 |
| $2t = 8$ | **96.2** |
| $2t = 10$ | 95.9 |

# B   TRAINING COMPARISONS BETWEEN SPOGW AND SCOREFLOW

As both our method and ScoreFlow are training-based approaches, we focus our comparison primarily on the ScoreFlow baseline. Optimization frameworks involving training generally exhibit more

Table 7: Impact of the dataset size $d$ on HumanEval performance. Optimal performance is achieved at $d = 100$, with degradation observed at both smaller and larger sizes, demonstrating the importance of balanced dataset scaling for effective policy optimization.

| The Number of Datapoints $d$ | HumanEval |
|---|---|
| $d = 50$ | 95.4 |
| $d = 100$ | **96.2** |
| $d = 200$ | 95.9 |
| $d = 300$ | 95.7 |
| $d = 400$ | 95.2 |
| $d = 500$ | 94.7 |
| $d = 600$ | 94.1 |

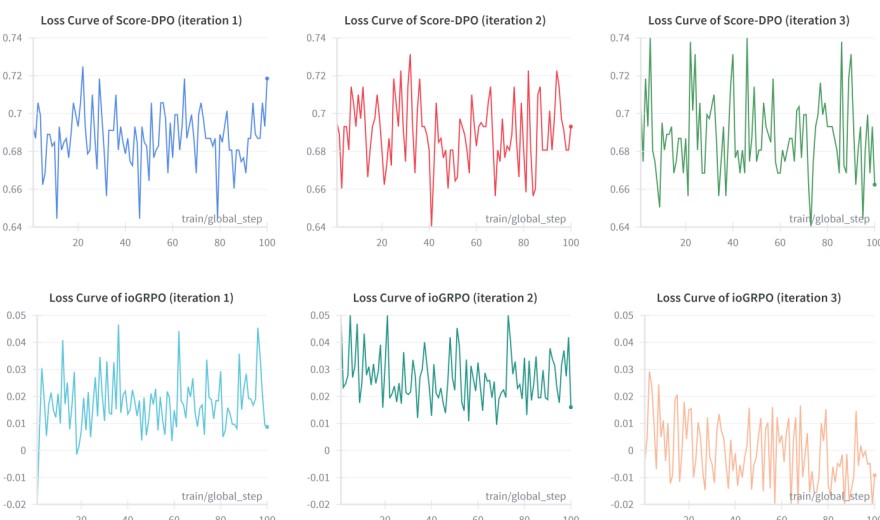

Figure 4: Comparisons of training loss curves between ioGRPO and Score-DPO on HumanEval.

stable and less variable time consumption across repeated runs, making them more comparable. We compared the total training optimization time and the final evaluation time for SPOGW and Score-Flow over three iterations on the HumanEval dataset, along with their respective training loss curves per iteration.

As shown in Table 8, the evaluation times for both methods are similar. During the optimization phase, SPOGW requires slightly more time than ScoreFlow due to its group-wise data processing and training, although the difference is marginal.

The training loss curves shown in Figure 4 reveal that throughout the training process (iterations 1-3), the Score-DPO loss values are generally higher and exhibit more pronounced oscillations, indicating weaker convergence. In contrast, the ioGRPO training process is more stable and demonstrates an overall trend of converging to lower values. This highlights the advantage of our method's training strategy in terms of stability and convergence behavior.

Table 8: Comparisons of the total training optimization time and the final evaluation time between SPOGW and ScoreFlow.

| Method | Optimization(h) | Evaluation(h) |
|---|---|---|
| ScoreFlow | 1.22 | 0.18 |
| **SPOGW** | **1.31** | **0.17** |

## C ADDITIONAL ABLATION STUDIES

### C.1 DISCUSSION ON THE CHARACTERISTICS OF IOGRPO

As discussed in Section 3.2, our Iterative Offline GRPO (ioGRPO) is modified from the standard online GRPO algorithm. It incorporates a key characteristic from offline RL—the decoupling of data generation from policy optimization—specifically to address the instability challenge inherent in workflow optimization tasks, thereby enabling GRPO's application in this domain. However, it is not a pure offline RL method. After each training epoch, the new model checkpoint interacts with the environment (i.e., the problem dataset) to generate and collect new training data (group preference data) before commencing the next epoch. As Levine et al. (2020) discussed, in pure offline RL, "the dataset is collected once, and is not altered during training; the training process does not interact with the MDP at all, and the policy is only deployed after being fully trained." Viewed in isolation per epoch, our training uses a static, offline dataset. However, the overall iterative process involves the new policy interacting with the environment and dynamically updating the training data, aligning with online RL characteristics. Precisely due to this hybrid nature, we term it Iterative Offline GRPO, distinguishing it from pure Offline GRPO.

Based on this discussion, we designed corresponding ablation experiments to compare pure offline GRPO, ioGRPO, and a variant of ioGRPO with more frequent environment interaction within epochs. Specifically, under identical experimental settings on HumanEval:

- **For offline GRPO**, group preference data was collected only in the first epoch and reused in all subsequent epochs without refresh.
- **For the ioGRPO (1/2 split) variant**, aiming for intra-epoch interaction, the problem dataset was split into two halves. The model interacted with and was trained on the first half, then interacted with and trained on the second half. (While standard online GRPO could partition data via batch size adjustment, this approach is infeasible for workflow optimization due to the instability introduced by execution, which was a key motivation for designing ioGRPO.)

The experimental results are shown in Table 9. As the results indicate, offline GRPO, requiring data collection only in the first round, has a relatively shorter average training time per epoch. However, it suffers from performance degradation in later epochs due to issues like out-of-distribution actions or data shift. In contrast, ioGRPO effectively mitigates these problems through the interaction of the new policy with the environment and the dynamic update of training data in each iteration, albeit at the cost of increased training time. Furthermore, we observe that the further partitioned variant increased training time without yielding a clear performance improvement.

Table 9: Comparisons of performance and average time among offline GRPO, ioGRPO and a variant of ioGRPO on HumanEval.

| Method | Epoch 1 | Epoch 2 | Epoch 3 | Epoch 4 | Avg Time per Epoch |
|---|---|---|---|---|---|
| offline GRPO | 95.4 | 95.2 | 94.7 | 93.9 | 20.3 min |
| **ioGRPO** | **95.4** | **96.2** | **96.2** | **95.2** | **36.2 min** |
| ioGRPO(1/2 split) | 94.1 | 95.2 | 95.9 | 94.9 | 41.3 min |

### C.2 GROUP-WISE AND PAIRWISE STRATEGY COMPARISON

To validate that the group-wise approach is superior to the pairwise paradigm, we used the score-integrated pairwise objective, Score-DPO, proposed in ScoreFlow as the baseline. This baseline utilizes the same data curation pipeline as our method to collect pairwise data, while SPOGW maintains its original group-wise data processing flow. Regarding the advantage-masked KL (mKL), it is challenging to deploy it identically in the Score-DPO baseline. This is because the DPO objective does not contain an explicit KL restriction term; instead, the KL restriction is implicitly embedded into its objective function via derivation (Rafailov et al., 2023), a formulation that Score-DPO inherits and does not alter. Therefore, for a fair comparison, we did not apply the KL mask to either method in this specific experiment. The results on the HumanEval benchmark are shown in Table 10,

where the group-wise optimization method (ioGRPO) still demonstrates superior performance. This highlights the inherent advantage of the group-wise paradigm.

Table 10: Comparison of the performance of ioGRPO and Score-DPO as the objective function for training respectively on the HumanEval benchmark.

| Method | HumanEval |
|---|---|
| Score-DPO | 94.9 |
| **ioGRPO** | **95.4** |

## C.3 EVALUATION ROBUSTNESS WITH DIFFERENT JUDGES

In our experiments on the HotpotQA, DROP, and MATH datasets, we used `GPT-4o-mini-2024-07-18` for both reward generation during training and final evaluation. Using the same model as the judge raises concerns about potential bias and "judge-specific overfitting".

To address this concern, we conducted an ablation study on the HotpotQA dataset, using ScoreFlow as the baseline. While maintaining `GPT-4o-mini-2024-07-18` for reward generation during training, we introduced different models—`DeepSeek-V3` and `GPT-5-nano-2025-08-07`—as independent judges during the final evaluation phase.

As the results shown in Table 11 demonstrate, while the absolute scores from different judges show minor variations, SPOGW consistently outperforms the baseline across all judge models. This consistency under different evaluation protocols strengthens the claim that our method learns to generate objectively better workflows rather than merely overfitting to the specific preferences of a single judge.

Table 11: Performance comparison of SPOGW and ScoreFlow under the final evaluation of different judges on HotpotQA.

| Method | GPT-4o-mini | DeepSeek-V3 | GPT-5-nano |
|---|---|---|---|
| ScoreFlow | 84.1 | 83.7 | 83.9 |
| **SPOGW** | **85.0** | **84.5** | **85.0** |

## D COST ANALYSIS OF SPOGW

The primary expenses in agentic workflow optimization stem from API calls for data collection and final evaluation. Below we detail the average cost per iteration for training data collection and a single inference test on the test set for SPOGW across five benchmarks in Table 12, using `GPT-4o-mini-2024-07-18` as the executor.

Table 12: The average cost per iteration for training data collection (Data Coll.) and a single inference test on the test set (Test) for SPOGW across five benchmarks.

| Benchmark | Data Coll. (USD) | Test(USD) |
|---|---|---|
| HumanEval | 0.96 | 0.23 |
| MBPP | 2.51 | 0.59 |
| HotpotQA | 6.05 | 1.41 |
| DROP | 5.84 | 1.39 |
| MATH | 4.30 | 1.02 |

Furthermore, we compared the total cost during the entire optimization phase on HumanEval against the non-training method AFlow and the training-based method ScoreFlow, also using `GPT-4o-mini-2024-07-18` as the executor. The results are shown in Table 13. Compared

to the non-training method, utilizing an open-source LLM as the foundation model and leveraging rapid convergence during training helps minimize costs. While our method incurs a moderate cost increase compared to ScoreFlow due to processing group-wise data, it remains significantly lower than AFlow.

Table 13: Comparison of the total cost during the entire optimization phase (Opt. Cost) of SPOGW, ScoreFlow, and AFlow on HumanEval.

| Method | Opt. Cost (USD) |
|---|---|
| AFlow | 4.65 |
| ScoreFlow | 2.40 |
| SPOGW | 2.88 |

Additionally, performance sensitivity to hyperparameter settings is a potential characteristic of various optimization methods. For SPOGW, our systematic ablation studies have identified a set of well-performing default hyperparameters. For new tasks, while the optimal values might be task-dependent, the tuning efforts and parameter ranges established in this paper provide a valuable reference point and a strong starting point for rapid hyperparameter tuning.

It is also important to note that the choice of executor in the SPOGW framework is not fixed. Exploring the performance-cost trade-off by utilizing cheaper APIs or powerful, locally deployable open-source models (e.g., `Qwen2.5-72B`, `Qwen3-235B-A22B`) as executors presents a viable strategy for further reducing operational expenses.

