# OpenReview forum: "SPOGW: a Score-based Preference Optimization method via Group-Wise comparison for workflows"
_ICLR.cc/2026/Conference — Submitted to ICLR 2026_

### Official Review · Reviewer_Nh7L · 2025-10-31

**Soundness:** 2
**Presentation:** 2
**Contribution:** 2
**Rating:** 2
**Confidence:** 3

**Summary:**

In this paper, the authors introduce SPOGW: a framework for automated optimization of agentic workflows for Large Language Models. Existing approaches are often constrained to be discrete via discrete optimization or pairwise preference modes such as DPO. Specifically, SPOGW proposes optimizing a generator LLM in a continuous space by means of a score-based, group-wise preference optimization approach.

The pipeline is composed of three main stages:
Firstly, a data generation and curation process. For each query, $m=16$ candidate workflows are generated and executed, and scalar reward scores are obtained. These are then aggregated into groups of size $n=14$ and subsequently filtered, screened, and selected for high intra-group reward variance. Finally, a "group sharpening" step reduces these groups by only retaining the top-t and bottom-t scoring responses;
Second, Iterative Offline Group Relative Policy Optimization is used. It combines the orthogonal strengths of aspirations of sample diversity with a low-variance, off-policy training procedure. The method decouples the inherently noisy and unstable data collection phase from policy update, and instead trains on a static, pre-collected dataset;
Thirdly, they use Advantage-Masked KL Restriction mKL, a modification of the regularized RL objective. The final loss selectively applies the KL-divergence penalty only to responses i for which the advantage is positive with a mask.

Empirically, SPOGW exhibits state-of-the-art performance across five benchmarks including mathematical reasoning, coding, and question answering, outperforming approaches including the DPO-based ScoreFlow.

**Strengths:**

1. The authors identify that online RL is challenging for the application of RL on agentic systems. By decoupling data collection from policy optimization, their ioGRPO transforms it into a stable, standard supervised-like training loop. This is a procedure that helps to alleviate the data staleness and distribution shift problems.
2. mKL better reconciles exploration and exploitation than a uniform KL penalty, which can create conflicting gradient signals by pushing the policy away from a bad action (via the PPO objective) while pulling it back (via the KL term). The ablation study in Table 3 provide validation for this insight.
3. They comment that the core of GRPO is the advantage calculation, which is highly sensitive to the reward distribution within a group. By selecting for groups with high variance and then sharpening them to further increase the spread between the best and worst responses, it ensures that the resulting advantage signals are strong and unambiguous. The ablation in Table 4 shows that this curation strategy is effective.
4. SPOGW can elevate smaller models to the performance level of much larger ones. The results show that a 3B-parameter Qwen2.5 model (optimized with SPOGW) nearly matches the performance of the baseline 14B model on HumanEval. It indicates that for complex and multi-step tasks, the reasoning process can be effectively factored into a smaller, more efficient generator model and an optimized workflow structure.

**Weaknesses:**

1. You mention that you generate $m=16$ candidate workflows for each problem, and that the optimal dataset is $d=100$ problems; this is done over 3 iterations. So if I am not mistaken, a simple calculation for a single benchmark’s training run is $100 \times 16 \times 3 = 4800$ workflow generations and executions, all using the proprietary GPT-4o-mini as the executor. The costs of such work should not be neglected.
2. The novelty of “Iterative offline GRPO” is actually contained within the already-established field of offline RL. While the iterative part that you refer to as “the first”, the most important part (the core) is what defines offline RL: data generation and policy optimizations are performed independently. I think your paper has too few citations and comparisons with the some offline RL methos (e.g., IQL, CQL). You should clarify whether your method solves the central problem of offline RL such as out-of-distribution actions and data shift, beyond the implicit regularization provided by the KL term.
3. The method relies on a single, powerful, proprietary model (GPT-4o-mini) as both the executor for reward generation and the judge for final evaluation. This setup may create a risk of "judge-specific overfitting." The generator model (Qwen2.5-7B) may not be learning to produce objectively better workflows, but rather workflows that are tailored to the specific biases of the GPT-4o-mini judge. The reported performance might not transfer if a different judge were used. The work would be much stronger if it demonstrated that the optimized workflows are superior under a different evaluation protocol.
4. In section 3.1 and Figure 1 there’s a “Filtering” step right before “Screening”. Any details about this? Is it filtering based on syntax, length, or some other settings?

**Questions:**

1. (related to weakness2) Any discussion on how your method compares to canonical offline RL methods like Conservative Q-Learning (CQL) or Implicit Q-Learning (IQL). Did you consider these algorithms as alternatives during your research?
2. Please refer to weakness 3.
3. Did you conduct any ablation study on isolating the contribution of group-wise GRPO from the rest of the components? To justify the argument that group-wise is better than pairwise, I would want to look at a baseline that uses a pairwise objective but also has your data curation pipeline and advantage-masked KL. How does SPOGW compare to it?
4. On the $C(m, n)$ permutations, do you just generate all $C(m, n)$ groups per query, or do you use a sampling strategy to knock this number down? If so, what’s your approach?
5. see weakness 4.

---

> ### Author Response · Authors · 2025-11-24
> **Response to Reviewer Nh7L [1/4]**
>
> We appreciate the reviewer for the thorough and detailed review, and for acknowledging the strengths of our method, including the stabilization via ioGRPO, the design of mKL, the data curation strategy, and the ability to enhance smaller models. Below, we provide point-by-point responses to the weaknesses and questions raised.
>
> **Response to Weakness 1:**
>
> > You mention that you generate $m=16$ candidate workflows for each problem, and that the optimal dataset is $d=100$ problems; this is done over 3 iterations. So if I am not mistaken, a simple calculation for a single benchmark's training run is $100 \\times 16 \\times 3=4800$ workflow generations and executions, all using the proprietary GPT-4o-mini as the executor. The costs of such work should not be neglected.
>
> We appreciate the reviewer's pertinent concern regarding the computational cost of our method.
>
> The primary expenses in agentic workflow optimization stem from API calls for data collection and final evaluation. We have detailed the average cost per iteration for training data collection and a single inference test across five benchmarks, using GPT-4o-mini-2024-07-18 as the executor:
>
> | Benchmark | Data Coll. (USD) | Test(USD) |
> |-----------|------------------|-----------|
> | HumanEval | 0.96             | 0.23      |
> | MBPP      | 2.51             | 0.59      |
> | HotpotQA  | 6.05             | 1.41      |
> | DROP      | 5.84             | 1.39      |
> | MATH      | 4.30             | 1.02      |
>
> Furthermore, we compared the total optimization cost on HumanEval against the non-training method AFlow and the training-based method ScoreFlow, maintaining GPT-4o-mini-2024-07-18 as the executor:
>
> | Method    | Opt. Cost (USD) |
> |-----------|-----------------|
> | AFlow     | 4.65            |
> | ScoreFlow | 2.40            |
> | SPOGW     | 2.88            |
>
> Compared to the non-training method, utilizing an open-source LLM as the base model and leveraging fast convergence during training helps minimize costs. While our method incurs a moderate cost increase compared to ScoreFlow due to processing group-wise data, it remains substantially more cost-effective than AFlow.
>
> It is also noteworthy that the choice of executor in the SPOGW framework is not fixed. Exploring the performance-cost trade-off by utilizing cheaper APIs or powerful, locally deployable open-source models (e.g., Qwen2.5-72B, Qwen3-235B-A22B) as executors presents a viable strategy for further reducing operational expenses.
>
> We believe these observations and cost analyses may mitigate the concern regarding the practical deployment of SPOGW. We will incorporate this discussion into the revised version. We appreciate the reviewer for the valuable suggestion.

---

> ### Author Response · Authors · 2025-11-24
> **Response to Reviewer Nh7L [2/4]**
>
> **Response to Weakness 2 & Question 1:**
>
> > The novelty of "Iterative offline GRPO" is actually contained within the already-established field of offline RL... I think your paper has too few citations and comparisons with the some offline RL methos (e.g., IQL, CQL). You should clarify whether your method solves the central problem of offline RL...
>
> > (Related Question) Any discussion on how your method compares to canonical offline RL methods like Conservative Q-Learning (CQL) or Implicit Q-Learning (IQL). Did you consider these algorithms as alternatives during your research?
>
> We appreciate the reviewer for the insightful question. Our Iterative Offline GRPO (ioGRPO) is modified from the standard online GRPO algorithm. It incorporates a key characteristic from offline RL---the decoupling of data generation from policy optimization---specifically to address the instability challenge inherent in workflow optimization tasks, thereby enabling GRPO's application in this domain. However, it is not a pure offline RL method. After each training epoch, the new model checkpoint interacts with the environment (i.e., the problem dataset) to generate and collect new training data (group preference data) before commencing the next epoch. As discussed in [1], in pure offline RL, "the dataset is collected once, and is not altered during training; the training process does not interact with the MDP at all, and the policy is only deployed after being fully trained." Viewed in isolation per epoch, our training uses a static, offline dataset. However, the overall iterative process involves the new policy interacting with the environment and dynamically updating the training data, aligning with online RL characteristics. Precisely due to this hybrid nature, we term it Iterative Offline GRPO, distinguishing it from pure Offline GRPO.
>
> Based on this discussion, we designed corresponding ablation experiments to compare pure offline GRPO, ioGRPO, and a variant of ioGRPO with more frequent environment interaction within epochs. Specifically, under identical experimental settings on HumanEval:
>
> - For offline GRPO, group preference data was collected only in the first epoch and reused in all subsequent epochs without refresh.
> - For the ioGRPO (1/2 split) variant, aiming for intra-epoch interaction, the problem dataset was split into two halves. The model interacted with and was trained on the first half, then interacted with and trained on the second half. (While standard online GRPO could partition data via batch size adjustment, this approach is infeasible for workflow optimization due to the instability introduced by execution, which was a key motivation for designing ioGRPO).
>
> The experimental results are shown in the table below:
>
> | Method             | Epoch 1 | Epoch 2 | Epoch 3 | Epoch 4 | Avg Time per Epoch |
> |--------------------|---------|---------|---------|---------|-------------------|
> | offline GRPO       | 95.4    | 95.2    | 94.7    | 93.9    | 20.3min           |
> | ioGRPO             | 95.4    | 96.2    | 96.2    | 95.2    | 36.2min           |
> | ioGRPO(1/2 split)  | 94.1    | 95.2    | 95.9    | 94.9    | 41.3min           |
>
> As the results indicate, offline GRPO, requiring data collection only in the first round, has a relatively shorter average training time per epoch. However, it suffers from performance degradation in later epochs due to issues like out-of-distribution actions or data shift. In contrast, ioGRPO effectively mitigates these problems through the interaction of the new policy with the environment and the dynamic update of training data in each iteration, albeit at the cost of increased training time. Furthermore, we observe that the further partitioned variant increased training time without yielding a clear performance improvement.
>
> Moreover, compared to algorithms like IQL, CQL, or PPO, the core advantage of GRPO lies in its fundamental paradigm shift: it abandons the need to learn a precise absolute value function. Instead, it achieves more stable and robust learning through relative comparisons within a group of samples. By focusing on the relative advantage of each action compared to others in the group, GRPO effectively circumvents the challenges associated with value function estimation bias.
>
> We will incorporate a discussion of these analyses and our method's specific contributions and positioning in the revised version. We appreciate the reviewer for the valuable question.
>
> [1] Levine, S., Kumar, A., Tucker, G., & Fu, J. (2020). Offline reinforcement learning: Tutorial, review, and perspectives on open problems. arXiv preprint arXiv:2005.01643.

---

> ### Author Response · Authors · 2025-11-24
> **Response to Reviewer Nh7L [3/4]**
>
> **Response to Weakness 3 & Question 2:**
>
> > The method relies on a single, powerful, proprietary model (GPT-4o-mini) as both the executor for reward generation and the judge for final evaluation. This setup may create a risk of "judge-specific overfitting."... The work would be much stronger if it demonstrated that the optimized workflows are superior under a different evaluation protocol.
>
> We appreciate the reviewer raising this critical point concerning evaluation robustness and potential judge-specific overfitting.
>
> In our experiments on the HotpotQA, DROP, and MATH datasets, we used GPT-4o-mini-2024-07-18 for both reward generation during training and final evaluation. Using the same model as the judge raises concerns about potential bias and "judge-specific overfitting."
>
> To address this concern, we conducted an ablation study on the HotpotQA dataset, using ScoreFlow as the baseline. While retaining GPT-4o-mini-2024-07-18 for reward generation during training, we introduced different models---DeepSeek-V3 and GPT-5-nano-2025-08-07---as independent judges during the final evaluation phase. The results are presented below:
>
> | Method    | GPT-4o-mini | DeepSeek-V3 | GPT-5-nano |
> |-----------|-------------|-------------|------------|
> | ScoreFlow | 84.1        | 83.7        | 83.9       |
> | SPOGW     | 85.0        | 84.5        | 85.0       |
>
> As the results demonstrate, while the absolute scores from different judges show minor variations, SPOGW consistently outperforms the baseline across all judge models. This consistency under different evaluation protocols strengthens the claim that our method learns to generate objectively better workflows rather than merely overfitting to the specific preferences of a single judge. We will incorporate this discussion into the revised version. We appreciate the reviewer for the valuable suggestion.
>
> **Response to Weakness 4 & Question 5:**
>
> > In section 3.1 and Figure 1 there's a "Filtering" step right before "Screening". Any details about this? Is it filtering based on syntax, length, or some other settings?
>
> We appreciate the reviewer pointing out the need for clarification on the "Filtering" step.
>
> The filtering process checks whether all workflows associated with a given query in a training instance have identical scores. If they are all the same, the entire instance is discarded; otherwise, it is retained. This is because when all scores within a group are identical, it becomes impossible to learn any meaningful relative advantage signals, which is detrimental to effective training. We have also included an experimental analysis of this filtering step in our data processing ablation studies.
>
> We will add these details to the revised version for better reproducibility and clarity.

---

> > ### Author Response · Authors · 2025-11-24
> > **Response to Reviewer Nh7L [4/4]**
> >
> > **Response to Question 3:**
> >
> > > Did you conduct any ablation study on isolating the contribution of group-wise GRPO from the rest of the components? To justify the argument that group-wise is better than pairwise, I would want to look at a baseline that uses a pairwise objective but also has your data curation pipeline and advantage-masked KL. How does SPOGW compare to it?
> >
> > We appreciate the valuable suggestion for ablating the group-wise objective against a strong pairwise baseline.
> >
> > We performed the following ablation study to validate that the group-wise approach is superior to the pairwise paradigm. We used the score-integrated pairwise objective, Score-DPO, proposed in ScoreFlow as the baseline. This baseline utilizes the same data curation pipeline as our method to collect pairwise data, while SPOGW maintains its original group-wise data processing flow. Regarding the advantage-masked KL (mKL), it is challenging to deploy it identically in the Score-DPO baseline. This is because the DPO objective does not contain an explicit KL restriction term; instead, the KL restriction is implicitly embedded into its objective function via derivation [2], a formulation that Score-DPO inherits and does not alter. Therefore, for a fair comparison, we did not apply the KL mask to either method in this specific experiment. The results on the HumanEval benchmark are shown below, where the group-wise optimization method (ioGRPO) still demonstrates superior performance.
> >
> > | Method     | HumanEval |
> > |------------|-----------|
> > | Score-DPO  | 94.9      |
> > | IoGRPO     | 95.4      |
> >
> > We believe this comparison effectively highlights the inherent advantage of the group-wise paradigm. We will incorporate this discussion into the revised version. We appreciate the reviewer for the valuable suggestion.
> >
> > [2] Rafailov, R., Sharma, A., Mitchell, E., Manning, C. D., Ermon, S., & Finn, C. (2023). Direct preference optimization: Your language model is secretly a reward model. Advances in neural information processing systems, 36, 53728-53741.
> >
> > **Response to Question 4:**
> >
> > > On the $C(m,n)$ permutations, do you just generate all $C(m,n)$ groups per query, or do you use a sampling strategy to knock this number down? If so, what's your approach?
> >
> > We appreciate the reviewer for this question regarding the computational practicality of group formation.
> >
> > In our actual implementation, the process of generating combinations is integrated with the filtering step. Each candidate group-wise data instance generated via the $C(m,n)$ combinations is immediately subjected to the filtering criterion to determine whether it should be discarded. This results in a filtered dataset that is subsequently used for further data processing steps.
> >
> > We will clarify the strategy in the revised version.
> >
> > **Final Note:**
> >
> > We deeply appreciate the reviewer's extensive and constructive engagement with our work. We have addressed the points regarding cost, novelty and relation to offline RL, evaluation robustness, methodological details, group-wise  against pairwise ablation studies, and computational implementation. We hope our responses have alleviated your concerns and kindly request you to consider a more favorable evaluation.

---

### Official Review · Reviewer_S45m · 2025-10-31

**Soundness:** 3
**Presentation:** 3
**Contribution:** 3
**Rating:** 6
**Confidence:** 3

**Summary:**

This paper addresses the optimization of "agentic workflows" for Large Language Models (LLMs) by proposing a new method called SPOGW (Score-based Preference Optimization via Group-Wise comparison). Currently, designing effective workflows requires significant manual effort, and existing automated methods are limited by discrete optimization, rigid pairwise comparison paradigms (like DPO), or unstable on-the-fly execution

**Strengths:**

First, applying GRPO (a non-pairwise RL algorithm) to the problem of agentic workflow optimization is novel. Second, adapting it into an ioGRPO (iterative offline framework) is a clever engineering solution that directly tackles the critical stability challenges faced by online RL methods in this domain (which requires execution) . Finally, the mKL is a very insightful contribution; it recognizes the flaw in the standard KL penalty (i.e., penalizing divergence from bad samples) and provides an elegant, targeted solution to only preserve the "good" parts of the reference model.

**Weaknesses:**

1. The hyperparameter search and the data collection itself may be expensive.  The ablation studies show that SPOGW's performance is notably sensitive to several key hyperparameters: the KL coefficient $\beta$, the group size $2t$, and the dataset size $d$. For example, performance on HumanEval peaks at 96.2 with $d=100$ but drops to 94.1 at $d=600$ 12; it is 96.2 for $\beta=0.1$ but drops to 94.7 for $\beta=0.025$. While the paper identifies the optimal values, it doesn't discuss the cost of finding these optimal settings on a new task.

2. Confusing "Offline" Terminology: The paper terms the method "Iterative offline GRPO". In reinforcement learning literature, "Offline RL" (or Batch RL) typically refers to learning only from a fixed, pre-collected dataset without further interaction with the environment. SPOGW's framework, however, is iterative, where the policy is used to collect new data after each iteration.

**Questions:**

Please see above。

---

> ### Author Response · Authors · 2025-11-24
> **Response to Reviewer S45m**
>
> We appreciate the reviewer for the encouraging feedback and for recognizing the novelty of applying GRPO to agentic workflows, the engineering solution of ioGRPO, and the insightfulness of the advantage-masked KL restriction. We address the two main concerns below.
>
> **Response to Weakness/Question 1:**
>
> > The hyperparameter search and the data collection itself may be expensive. The ablation studies show that SPOGW's performance is notably sensitive to several key hyperparameters... While the paper identifies the optimal values, it doesn't discuss the cost of finding these optimal settings on a new task.
>
> We appreciate the reviewer's pertinent concern regarding the computational cost and hyperparameter sensitivity.
>
> The primary expenses in agentic workflow optimization stem from API calls for data collection and final evaluation. Below we detail the average cost per iteration for training data collection and a single inference test on the test set for SPOGW across five benchmarks, using GPT-4o-mini-2024-07-18 as the executor:
>
> | Benchmark | Data Coll. (USD) | Test(USD) |
> |-----------|------------------|-----------|
> | HumanEval | 0.96             | 0.23      |
> | MBPP      | 2.51             | 0.59      |
> | HotpotQA  | 6.05             | 1.41      |
> | DROP      | 5.84             | 1.39      |
> | MATH      | 4.30             | 1.02      |
>
> Furthermore, we compared the total cost during the entire optimization phase on HumanEval against the non-training method AFlow and the training-based method ScoreFlow, also using GPT-4o-mini-2024-07-18 as the executor:
>
> | Method    | Opt. Cost (USD) |
> |-----------|-----------------|
> | AFlow     | 4.65            |
> | ScoreFlow | 2.40            |
> | SPOGW     | 2.88            |
>
> Compared to the non-training method, using an open-source LLM as the base model and leveraging fast convergence during training helps minimize costs. While our method incurs a slightly higher cost than ScoreFlow due to processing group-wise data instead of pairwise data, it remains significantly lower than AFlow.
>
> Additionally, performance sensitivity to hyperparameter settings is a potential characteristic of various optimization methods. For SPOGW, our systematic ablation studies have identified a set of well-performing default hyperparameters. For new tasks, while the optimal values might be task-dependent, the tuning efforts and parameter ranges established in this paper provide a valuable reference point and a strong starting point for rapid hyperparameter tuning.
>
> It is also important to note that the choice of executor in the SPOGW framework is not fixed. Exploring the performance-cost trade-off by utilizing cheaper APIs or powerful, locally deployable open-source models (e.g., Qwen2.5-72B, Qwen3-235B-A22B) as executors presents a viable strategy for further reducing operational expenses.
>
> We believe these observations and cost analyses may mitigate the concern regarding the practical deployment of SPOGW. We will incorporate this discussion into the revised version. We appreciate the reviewer for the valuable suggestion.
>
> **Response to Weakness/Question 2:**
>
> > Confusing "Offline" Terminology: The paper terms the method "Iterative offline GRPO". In reinforcement learning literature, "Offline RL" (or Batch RL) typically refers to learning only from a fixed, pre-collected dataset without further interaction with the environment. SPOGW's framework, however, is iterative, where the policy is used to collect new data after each iteration.
>
> We appreciate the reviewer for this insightful question and the important clarification regarding terminology. Indeed, Iterative Offline GRPO (ioGRPO) is not a pure offline reinforcement learning method. However, compared to standard online reinforcement learning, ioGRPO incorporates a key characteristic from offline RL---the decoupling of data generation from policy optimization---specifically to address the instability challenge inherent in workflow optimization tasks. If we examine each epoch in isolation, training is performed on a static, offline dataset. Yet, when viewing the entire iterative training process, each iteration involves the new policy model interacting with the environment to dynamically refresh the training data, which aligns with the characteristics of online reinforcement learning. Precisely because of this hybrid nature, we refer to our method as Iterative Offline GRPO, explicitly distinguishing it from pure Offline GRPO. We will clarify this terminology in the revised version to prevent any potential confusion.
>
> **Final Note:**
>
> We are grateful for the reviewer's positive assessment and insightful suggestions regarding cost and terminology. We hope our clarifications have addressed your concerns. We would be honored to receive your continued support.

---

### Official Review · Reviewer_kNmS · 2025-11-02

**Soundness:** 3
**Presentation:** 2
**Contribution:** 3
**Rating:** 6
**Confidence:** 1

**Summary:**

This paper primarily focuses on automated agentic workflow generation. First, the authors propose a score-based preference optimization method that directly leverages cardinal reward signals for policy updates. This mitigates the inherent limitations of existing workflow optimization methods. The superior performance of the proposed method is demonstrated through experiments.

**Strengths:**

- The problem is well-motivated. Optimizing the agentic workflow is crucial for practical applications.
- The experiments are comprehensive, and the proposed method demonstrates better performance than prior work.

**Weaknesses:**

I am not an expert in automated agentic workflow generation, so I am not sure whether the experimental setup is fair. Specifically, I did not understand why the LLMs used for optimization and the generator models differ across the baselines and the proposed method in the experiments. What if the same LLM is used for all methods? Does the proposed method still perform better than the others?

Moreover, I am wondering whether the training time is comparable across all methods. Are there experimental results showing training curves with respect to GPU hours?

**Questions:**

My main concerns and questions are outlined in the Weaknesses section. Additionally, I have the following question:

- Eq. (4) is not equivalent to the original definition of the KL divergence. Does it actually denote the derivative of the KL divergence?

---

> ### Author Response · Authors · 2025-11-24
> **Response to Reviewer kNmS**
>
> We appreciate the reviewer for the positive assessment of our work's motivation and the comprehensiveness of our experiments. We are pleased that the reviewer recognizes the importance of optimizing agentic workflows. Below, we respond to each of the concerns raised.
>
> **Response to Weakness/Question 1:**
>
> > I am not an expert in automated agentic workflow generation, so I am not sure whether the experimental setup is fair. Specifically, I did not understand why the LLMs used for optimization and the generator models differ across the baselines and the proposed method in the experiments. What if the same LLM is used for all methods? Does the proposed method still perform better than the others?
>
> We appreciate the reviewer's question regarding the fairness of our experimental setup and the choice of LLMs.
>
> Among the baseline methods, unlike other optimization approaches that directly utilize powerful existing models for execution and optimization, the ScoreFlow framework incorporates model training. It fine-tunes a small open-source model to become a high-performance agentic workflow generator, which is then paired with an off-the-shelf executor to complete tasks. Therefore, while maintaining GPT-4o-mini-2024-07-18 as the consistently used model for direct inference components, Qwen2.5-7B-Instruct was designated as the open-source model to be trained within the ScoreFlow method. Since our approach similarly involves training an open-source model as the generator, we kept the model configurations aligned with ScoreFlow to ensure a fair comparison.
>
> We believe these clarifications affirm the fairness and robustness of our experimental comparisons.
>
> **Response to Weakness/Question 2:**
>
> > Moreover, I am wondering whether the training time is comparable across all methods. Are there experimental results showing training curves with respect to GPU hours?
>
> We appreciate the reviewer for raising this point about training efficiency and the suggestion for training curves.
>
> As both our method and ScoreFlow are training-based approaches, we focus our comparison primarily on the ScoreFlow baseline. Optimization frameworks involving training generally exhibit more stable and less variable time consumption across repeated runs, making them more comparable.
>
> We compared the total training optimization time and the final evaluation time for SPOGW and ScoreFlow over three iterations on the HumanEval dataset, along with their respective training loss curves per iteration(Figure 4 in Appendix B of the preliminary revised version).
>
> | Method    | Optimization(h) | Evaluation(h) |
> |-----------|-----------------|---------------|
> | ScoreFlow | 1.22            | 0.18          |
> | SPOGW     | 1.31            | 0.17          |
>
> As shown in the table, the evaluation times for both methods are similar. During the optimization phase, SPOGW requires slightly more time than ScoreFlow due to its group-wise data processing and training, although the difference is marginal.
>
> Analysis of the training loss curves reveals that throughout the training process (iterations 1-3), the Score-DPO loss values are generally higher and exhibit more pronounced oscillations, indicating weaker convergence. In contrast, the ioGRPO training process is more stable and demonstrates an overall trend of converging to lower values. This highlights the advantage of our method's training strategy in terms of stability and convergence behavior.
>
> We hope this addresses the reviewer's question regarding training efficiency and stability. We will incorporate this discussion into the revised version. We appreciate the reviewer for the valuable suggestion.
>
> **Response to Question 3:**
>
> > Eq. (4) is not equivalent to the original definition of the KL divergence. Does it actually denote the derivative of the KL divergence?
>
> We adopted the definition of the KL divergence as presented in the original GRPO paper [1], making no modifications ourselves. The authors of [1] proposed using this specific unbiased estimator to estimate the KL divergence, ensuring its value remains positive.
>
> [1] Shao, Z., Wang, P., Zhu, Q., Xu, R., Song, J., Bi, X., ... & Guo, D. (2024). Deepseekmath: Pushing the limits of mathematical reasoning in open language models. arXiv preprint arXiv:2402.03300.
>
> **Final Note:**
>
> We appreciate the reviewer's thoughtful questions and careful examination of our work. We hope our responses have clarified the experimental design, training efficiency comparisons, and mathematical details. We would be honored to receive your continued support.

---

### Official Review · Reviewer_vCiT · 2025-11-03

**Soundness:** 3
**Presentation:** 3
**Contribution:** 2
**Rating:** 4
**Confidence:** 3

**Summary:**

The paper presents SPOGW, a new method for optimizing agentic workflows (structured multi-step sequences of calls by large language models). The core innovations are:
- Constructing group-wise training data — for each input query, generate multiple candidate workflows, execute them to get scalar scores, then form groups of workflows and use the score distributions in the group to compute advantages for optimization.
- Using an Iterative Offline Group Relative Policy Optimization (ioGRPO) loop, where data collection (workflow generation + execution + scoring) is decoupled from policy updates to avoid code execution and API instability during training.
- Applying an advantage-masked KL divergence (mKL) regularizer: the KL penalty (between current policy and reference policy) is applied selectively only on responses with positive advantage, thus encouraging alignment to good behaviors while allowing learning beyond the reference.

**Strengths:**

- The overall paper is presented well, with the entire workflow and key ideas explained clearly. Especially, the steps for dataset collection, filtering, sharpening are illustrated well.

- The idea of applying Advantage-Masked KL Restriction is interesting. Intuitively, this is a good approach for preventing the policy updating too much for good actions, while allowing flexibility for bad actions.

- Several ablation studies are performed, which strengthen the paper a lot, especially, the ones on data preprocessing methods and KL regularization.

**Weaknesses:**

- The overall flow is largely based on the standard RLVR framework, with modifications added (i.e., offline and advantage KL restriction). It is unclear whether the proposed modifications are generalizable beyond the current setting and experiments (e.g., whether it works in math without the agentic workflows).

- The effectiveness of offline iterative GRPO is not well illustrated. Especially, offline should introduce a huge impact of off-policy, while improving training time. It would be more convincing if the benefits from shorter training time (and/or better stability) overwhelms the suffering from the off-policy impact. I would suggest adding some comparisons (both training time and final results) between the online and offline GRPO.

**Questions:**

My concerns are center on the two points listed in weakness. I would love to hear the authors' opinions on them.

---

> ### Author Response · Authors · 2025-11-24
> **Response to Reviewer vCiT [1/2]**
>
> We appreciate the reviewer for the thoughtful feedback and for recognizing the clarity of our presentation and the value of our ablation studies. We are pleased that the reviewer considered the Advantage-Masked KL restriction valuable. Below, we address the two main concerns raised.
>
> **Response to Weakness/Question 1:**
>
> > The overall flow is largely based on the standard RLVR framework, with modifications added (i.e., offline and advantage KL restriction). It is unclear whether the proposed modifications are generalizable beyond the current setting and experiments (e.g., whether it works in math without the agentic workflows).
>
> We appreciate the reviewer's insightful question regarding the generalizability of our proposed modifications.
>
> The modified training framework presented in our work was primarily designed for the agentic workflow optimization task, but its applicability is not strictly limited to this domain. The modifications, namely the Iterative Offline GRPO (ioGRPO) and the advantage-masked KL restriction, originated from adapting the standard online GRPO algorithm---commonly applicable to classic tasks like mathematical reasoning and question answering---to the specific challenges posed by optimizing workflows.
>
> Fundamentally, decoupling data collection from training and incorporating the advantage-masked KL restriction do not alter the core mechanics of GRPO: sampling, reward scoring, advantage calculation, and policy optimization. However, the unique nature of workflows introduces distinct challenges, particularly in the reward scoring phase. Unlike classic tasks where output quality can be directly assessed by a deterministic scoring function, workflows lack a straightforward scoring standard. Their quality must be empirically evaluated through actual execution. This necessitates an additional execution verification step prior to scoring, unlike in classic problems. This execution step introduces instability due to potential code execution failures and API call issues. To prevent this inherent instability from affecting the training process, ioGRPO decouples the data collection phase (which includes sampling, execution and reward scoring) from the training phase (which involves advantage calculation and policy optimization). The KL restriction was subsequently refined to suit this adapted framework.
>
> Consequently, from an implementation perspective, our training framework can theoretically be applied to classic tasks like math and QA, where data collection would involve sampling followed by direct scoring. While its performance across various tasks and a direct comparison with the original GRPO in those settings require further investigation, the framework presents a potentially viable solution or a source of inspiration for problem domains sharing similar characteristics with workflow optimization---specifically, those where the reward scoring process is complex and inherently unstable.
>
> We believe these clarifications demonstrate the broader applicability and conceptual contribution of our framework beyond the specific domain of agentic workflows.

---

> ### Author Response · Authors · 2025-11-24
> **Response to Reviewer vCiT [2/2]**
>
> **Response to Weakness/Question 2:**
>
> > The effectiveness of offline iterative GRPO is not well illustrated. Especially, offline should introduce a huge impact of off-policy, while improving training time. It would be more convincing if the benefits from shorter training time (and/or better stability) overwhelms the suffering from the off-policy impact. I would suggest adding some comparisons (both training time and final results) between the online and offline GRPO.
>
> We appreciate the reviewer for the valuable suggestion. We would like to clarify that our Iterative Offline GRPO (ioGRPO) is not a pure offline reinforcement learning method. After each training epoch, the new model checkpoint interacts with the environment (i.e., the problem dataset) to generate and collect new training data (the group preference data) before commencing the next epoch. As discussed in [1], in offline RL, "the dataset is collected once, and is not altered during training; the training process does not interact with the MDP at all, and the policy is only deployed after being fully trained." Viewed per epoch, training occurs on a static, offline dataset. However, considering the entire iterative process, each iteration involves the new policy interacting with the environment and dynamically refreshing the training data, which aligns with the characteristics of online reinforcement learning. Precisely for this reason, we term our method Iterative Offline GRPO, distinguishing it from pure Offline GRPO.
>
> Based on this discussion, we designed corresponding ablation experiments to compare pure offline GRPO, ioGRPO, and a variant of ioGRPO with more frequent environment interaction within epochs. Specifically, under identical experimental settings on HumanEval:
>
> - For offline GRPO, group preference data was collected only in the first epoch and reused in all subsequent epochs without refresh.
> - For the ioGRPO (1/2 split) variant, aiming for intra-epoch interaction, the problem dataset was split into two halves. The model interacted with and was trained on the first half, then interacted with and trained on the second half. (While standard online GRPO could partition data via batch size adjustment, this approach is infeasible for workflow optimization due to the instability introduced by execution, which was a key motivation for designing ioGRPO).
>
> The experimental results are shown in the table below:
>
> | Method             | Epoch 1 | Epoch 2 | Epoch 3 | Epoch 4 | Avg Time per Epoch |
> |--------------------|---------|---------|---------|---------|-------------------|
> | offline GRPO       | 95.4    | 95.2    | 94.7    | 93.9    | 20.3min           |
> | ioGRPO             | 95.4    | 96.2    | 96.2    | 95.2    | 36.2min           |
> | ioGRPO(1/2 split)  | 94.1    | 95.2    | 95.9    | 94.9    | 41.3min           |
>
> As the results indicate, offline GRPO, requiring data collection only in the first round, has a relatively shorter average training time per epoch. However, it suffers from performance degradation in later epochs due to issues like out-of-distribution actions or data shift. In contrast, ioGRPO effectively mitigates these problems through the interaction of the new policy with the environment and the dynamic update of training data in each iteration, albeit at the cost of increased training time. Furthermore, we observe that the further partitioned variant increased training time without yielding a clear performance improvement.
>
> We believe that these additional analyses and clarifications are able to substantiate the practical benefits and design rationale of our iterative, decoupled approach. We will incorporate this discussion into the revised version. We appreciate the reviewer for the valuable suggestion.
>
> [1] Levine, S., Kumar, A., Tucker, G., & Fu, J. (2020). Offline reinforcement learning: Tutorial, review, and perspectives on open problems. arXiv preprint arXiv:2005.01643.
>
> **Final Note:**
>
> We sincerely appreciate the reviewer for the constructive feedback, which has helped us improve the paper. We hope our responses have adequately addressed the concerns regarding generalizability and the effectiveness of ioGRPO. We kindly invite the reviewer to consider a more favorable evaluation.

---

### Author Response · Authors · 2025-11-28
**Modification Note**

Dear Reviewers,

We deeply appreciate your valuable feedback and constructive comments on our paper. We have carefully revised the manuscript based on your suggestions. The main text has been enhanced with new content (highlighted in red font for your convenience), and we have added comprehensive appendices with detailed discussions and experimental results.

## Response to Reviewer vCiT

**Weakness/Question 1: Generalizability of Proposed Modifications**
- Added a new subsection **"3.4 Discussion on Generalizability"** in the main text that explicitly discusses the broader applicability of our framework beyond agentic workflow optimization.

**Weakness/Question 2: Effectiveness of Iterative Offline GRPO**
- Enhanced Section 3.2 with clarification that ioGRPO is not a pure offline RL method.
- Added comprehensive ablation studies in **Appendix C.1** comparing pure offline GRPO, ioGRPO, and a variant.

## Response to Reviewer kNmS

**Weakness/Question 1: Fairness of Experimental Setup**
- Added clarification in Section 4.1 explaining consistent model configurations for fair comparison.

**Weakness/Question 2: Training Time Comparison**
- Added **Appendix B** with training time comparisons and loss curves between SPOGW and ScoreFlow.

**Weakness/Question 3: KL Restriction Definition**
- Added clarification in Section 3.3 noting that Equation (4) adopts the KL definition from original GRPO.

## Response to Reviewer S45m

**Weakness/Question 1: Computational Cost Analysis**
- Added comprehensive cost analysis in **Appendix D** with detailed cost breakdowns across benchmarks.

**Weakness/Question 2: "Offline" Terminology Clarification**
- Enhanced Section 3.2 with explicit clarification of ioGRPO's hybrid nature.

## Response to Reviewer Nh7L

**Weakness 1: Computational Cost Concerns**
- Integrated detailed cost analysis throughout the paper and in **Appendix D**.

**Weakness 2 & Question 1: Relation to Offline RL**
- Expanded Section 3.2 with discussion on ioGRPO's positioning.
- Added ablation studies comparing with pure offline GRPO in **Appendix C.1**.

**Weakness 3 & Question 2: Evaluation Robustness**
- Added **Appendix C.3** with evaluation using different judge models, demonstrating consistent superiority.

**Weakness 4 & Question 5: Filtering Step Details**
- Enhanced Section 3.1 with detailed explanation of filtering criteria.

**Question 3: Group-wise vs Pairwise Ablation**
- Added **Appendix C.2** with direct comparison validating the group-wise approach.

**Question 4: Computational Practicality**
- Enhanced Section 3.1 with clarification on implementation details.

We sincerely hope these comprehensive revisions have adequately addressed your concerns and questions. We look forward to your favorable assessment and extend our gratitude once again for your invaluable feedback, which has contributed to the enhancement of our work.

Sincerely,
The Authors

---

### Author Response · Authors · 2025-12-02
**Response to the Area Chair (Summary Statement) [1/2]**

Dear Area Chair,

We sincerely thank you for undertaking the evaluation of our submission, “SPOGW: a Score-based Preference Optimization method via Group-Wise comparison for workflows.” We also extend our gratitude to the original reviewers for their careful reading, constructive feedback, and detailed evaluations, which have greatly helped us improve the paper.

We respectfully provide this summary of our responses and corresponding revisions, along with the updated manuscript (the main text has been enhanced with new content, highlighted in blue font for your convenience) , to support your final assessment. Below we outline how we addressed each reviewer’s concerns, with specific references to changes in the revised manuscript.

---

**Reviewer vCiT**

**Weakness/Question 1 – Generalizability of modifications beyond workflow tasks**
We have added a dedicated subsection **3.4 Discussion on Generalizability** (pages 6–7) to explicitly discuss how the core components of SPOGW—Iterative Offline GRPO (ioGRPO) and advantage-masked KL restriction—can be conceptually extended to other domains, especially those where reward estimation is complex or unstable.

**Weakness/Question 2 – Effectiveness of Iterative Offline GRPO**
We have:
1. Clarified in **Section 3.2** (page 5) that ioGRPO is *not* a pure offline RL method, but a hybrid that decouples data collection from training updates for stability while retaining online-style iterative data refresh.
2. Provided an extensive ablation study in **Appendix C.1** (page 17, Table 9) comparing pure offline GRPO, ioGRPO, and a variant with intra‑epoch interaction. The results show that ioGRPO maintains high performance across epochs, avoiding the degradation observed with pure offline GRPO, at a moderate increase in training time.

---

**Reviewer kNmS**

**Weakness/Question 1 – Fairness of experimental setup (model choice)**
We have added a clarifying note in **Section 4.1** “Baselines” (page 7) explaining that, for fair comparison, we kept the same open‑source generator model (Qwen2.5‑7B‑Instruct) for both our method and ScoreFlow, as both are training‑based approaches.

**Weakness/Question 2 – Training comparison**
We have introduced **Appendix B** (pages 15–16) containing:
- Table 8: Comparisons of total optimization time and final evaluation time between SPOGW and ScoreFlow on HumanEval.
- Figure 4: Training loss curves illustrating the more stable convergence of ioGRPO compared to Score‑DPO.

**Question 3 – KL restriction definition**
We have added a sentence in **Section 3.3** (page 6) noting that Eq. (4) follows the KL‑restriction formulation from the original GRPO paper, which uses an unbiased estimator to ensure positivity.

---

**Reviewer S45m**

**Weakness/Question 1 – Cost of hyperparameter search and data collection**
We have added a comprehensive cost analysis in **Appendix D** (pages 18–19):
- Table 12: Per‑iteration data‑collection and test costs across five benchmarks.
- Table 13: Total optimization‑phase cost comparison with AFlow and ScoreFlow on HumanEval.

We also discuss strategies for cost reduction (e.g., using cheaper APIs or locally deployable models) and note that the hyperparameter tuning ranges provided in the paper offer a strong starting point for new tasks.

**Weakness/Question 2 – Clarification of “offline” terminology**
We have revised **Section 3.2** (page 5) to explicitly distinguish ioGRPO from pure offline RL, explaining that it combines offline‑style decoupled training with iterative online data collection.

---

---

> ### Author Response · Authors · 2025-12-02
> **Response to the Area Chair (Summary Statement) [2/2]**
>
> **Reviewer Nh7L**
>
> **Weakness 1 – Computational cost**
> The same cost analysis in **Appendix D** addresses this concern, showing that SPOGW remains significantly cheaper than non‑training baselines (e.g., AFlow) while being only moderately more expensive than ScoreFlow.
>
> **Weakness 2 & Question 1 – Relation to offline RL and comparison**
> We have:
> - Expanded the discussion in **Section 3.2** (page 5) on how ioGRPO differs from pure offline RL and why GRPO’s group‑relative advantage estimation circumvents value function estimation bias.
> - Added the ablation study in **Appendix C.1** (Table 9) demonstrating ioGRPO’s ability to mitigate data‑shift issues compared to pure offline GRPO.
>
> **Weakness 3 & Question 2 – Judge‑specific overfitting**
> We conducted an additional experiment using different judge models (DeepSeek‑V3, GPT‑5‑nano) for final evaluation on HotpotQA. The results, presented in **Appendix C.3** (page 18, Table 11), show that SPOGW consistently outperforms ScoreFlow across all judges, confirming robustness to judge‑specific biases.
>
> **Weakness 4 & Question 5 – Details of the “Filtering” step**
> We have elaborated **Section 3.1** (page 4) to explicitly describe the filtering criterion: instances where all workflows in a group have identical scores are discarded, as they provide no meaningful advantage signal.
>
> **Question 3 – Group‑wise vs. pairwise ablation**
> We added **Appendix C.2** (pages 17–18, Table 10) comparing group‑wise ioGRPO with the pairwise Score‑DPO objective (using the same data‑curation pipeline). The results confirm the superiority of the group‑wise approach (95.4 vs. 94.9 on HumanEval).
>
> **Question 4 – Computational practicality of group formation**
> We clarified in **Section 3.1** (page 4) that combination generation is interleaved with filtering, so only valid (non‑identical‑score) groups are retained.
>
> ---
>
> **Summary of Major Additions**
> The revised manuscript now includes:
> 1. **New subsection 3.4** on generalizability.
> 2. **Enhanced methodological clarifications** in Sections 3.1–3.3.
> 3. **Sufficient experiments with three new appendices** (B–D):
>    - Training‑time and loss‑curve comparisons (B).
>    - Ablations on ioGRPO variants, group‑wise vs. pairwise, and multi‑judge evaluation (C.1-3).
>    - Detailed cost analysis (D).
> 4. **Additional tables and figures** supporting all new experiments.
>
> We believe these revisions sufficiently address each concern raised by the reviewers and substantially strengthen the paper’s contribution, clarity, and empirical support.
>
> Once again, we sincerely thank you for your time and careful consideration of our work, and we also wish to express our appreciation to the original reviewers for their invaluable input. We respectfully submit this updated manuscript and summary for your consideration, and we look forward to your assessment.
>
> Sincerely,
> The Authors

---

### Meta-Review · Area_Chair_4ZE1 · 2026-01-05

**Summary:**

This paper was reviewed by four reviewers, with initial scores 4/6/6/2. The authors have provided detailed responses to the concerns and questions raised by the reviewers. While some of the concerns have been addressed, some concerns remain unaddressed. Specifically, one reviewer is concerned that the proposed KL Restriction is highly sensitive to hyperparameters. Also, the proposed ioGRPO that alternates data generation and training every epoch is similar to that of the data sync strategy already studied in ''Bridging Offline and Online Reinforcement Learning for LLMs''. The data filtering strategy, e.g., removing same-score or duplicate responses within groups, is similar to the “dynamic sample” in ''DAPO: An Open-Source LLM Reinforcement Learning System at Scale''. Combined with the reviewers' concerns regarding the technical contribution, the AC believes that more clarifications are needed to differentiate the proposed method from the existing designs. Given ICLR’s highly competitive acceptance rate this year, the AC regrets to recommend rejection.

**Reviewer Concerns:**

### Reviewer vCiT

1. Weakness 1: The author's method is based on the RLVR framework with incremental improvements, and it would be better if the discussion on the generalizability of the added KL restriction and its performance in non-agentic scenarios were provided.

2. Weakness 2: Though the authors have responded, a detailed comparison between online GRPO and offline GRPO in terms of training time, stability, and performance results would be more beneficial. The authors provided experimental results with offline GRPO but did not include direct comparisons with online GRPO.  Therefore, Reviewer vCiT might not raise the score based on the discussion.

### Reviewer kNmS

Regarding the inconsistency between the generation model and the optimization model, the author responded that the workflow generation uses fine-tuned high-performance models, while the settings of the proposed work is consistent with the settings of ScoreFlow.
On the other hand, As for the issue of training curves, the author provided additional training curves for Score-DPO and ioGRPO, demonstrating the stability of ioGRPO.

### Reviewer S45m

1. Weakness 1: The performance of this method may be sensitive to specific parameters and raises concerns about generalizability. The author responded that high sensitivity is commonly found in various optimization methods and that hyperparameter settings are task-dependent. However, the author's response lacks a analysis regarding the stability of the method. This might not address the reviewer's concerns.

2. Weakness 2: The author misused the term "offline." The author promised that this would be corrected in a revised version.


### Reviewer Nh7L
1. Weakness 1: The generation and execution of workflows rely on closed-source models, which induces cost. The author responded that the execution part can be replaced with open-source models, and while the cost is higher than that of the ScoreFlow method, it is lower than that of the Aflow method. The proposed method incurs a moderate cost increase compared to ScoreFlow.

2. Weakness 2: The improvements of ioGRPO have already been included in existing offline reinforcement learning methods, and there is a lack of comparison with other existing reinforcement methods. The author responded with experimental results comparing ioGRPO with offline GRPO, but this does not fully address the novelty issue. The pipeline of training for one epoch, then regenerating the data and proceeding with further training has already been explored in GRPO-based papers. On the other hand, the author addressed the differences between IQL, CQL, or PPO and GRPO, but incorporating an analysis regarding the difference between ioGRPO and GRPO, as well as other reinforcement learning algorithms, would be better.

3. Weakness 3: The potential bias issue due to the similarity between the generation model and the scoring model, the author conducted experiments using different scoring models, which yielded consistent results under different models, indicating that the improvements of this method are not affected by the model bias. This weakness has been addressed in the rebuttal.

4. Weakness 4: The reviewer concerns the need to generate C(m, n) groups and whether there are pruning strategies. The author responded that filtering has already been done during workflow generation.

**Reviewer Scores:**

The scores were 4/6/6/2, with confidence 3/1/3/3. The concerns of the reviewer kNmS with confidence 1 have been addressed. However, the score might not be representative, as the reviewer kNmS acknowledged that he/she was not familiar with the topic of this paper.

The reviewers vCiT and Nh7L gave 4 and 2, respectively. However, the AC has checked the rebuttal and found that several concerns were not well addressed. Therefore, the scores may be maintained.

Reviewer S45m gave a score of 6 and raised questions regarding the method’s stability. However, the rebuttal does not include a sensitivity analysis to address this, and the reviewer is unlikely to further increase the score.

---

### Decision · Program_Chairs · 2026-01-26

Reject